# Environmental controls on the elemental composition of a Southern Hemisphere strain of the coccolithophore *Emiliania huxleyi*

Yuanyuan Feng[1, 2, 3, 4], Michael Y. Roleda[2, 5], Evelyn Armstrong[6], Cliff S. Law[6, 7], Philip W. Boyd[6, 7, 8], Catriona L. Hurd[2, 8]

5 [1] College of Marine and Environmental Sciences, Tianjin University of Science and Technology, Tianjin 300457, China
[2] Department of Botany, University of Otago, PO Box 56, Dunedin 9054, New Zealand
[3] Tianjin Key Laboratory of Marine Resources and Chemistry, Tianjin 300457, China
[4] Tianjin Marine Environmental Protection and Restoration Technology Engineering Center, Tianjin 10 300457, China
[5] Norwegian Institute of Bioeconomy Research, Bodø 8027, Norway
[6] NIWA/University of Otago Research Centre for Oceanography, Department of Chemistry, University of Otago, PO Box 56, Dunedin 9054, New Zealand
[7] NIWA, Greta Point, Wellington 6023, New Zealand
15 [8] Institute for Marine and Antarctic Studies, University of Tasmania, Hobart, 7005, Tasmania, Australia

*Correspondence to*: Yuanyuan Feng (yfeng@tust.edu.cn)

**Abstract.** A series of semi-continuous incubation experiments were conducted with the coccolithophore *Emiliania huxleyi* strain NIWA1108 (Southern Ocean isolate) to examine the effects of five environmental drivers (nitrate and phosphate concentrations, irradiance, temperature, and partial 20 pressure of $CO_2$ ($pCO_2$)) on both the physiological rates and elemental composition of the coccolithophore. Here, we report the alteration of the elemental composition of *E. huxleyi* in response to the changes in these environmental drivers. A series of dose response curves for the cellular elemental composition of *E. huxleyi* were fitted for each of the five drivers across an environmentally-representative gradient. The importance of each driver in regulating the elemental composition of *E. 25 huxleyi* was ranked using a semi-quantitative approach. The percentage variations in elemental composition arising from the change in each driver between present day and model-projected conditions for the year 2100 were calculated. Temperature was the most important driver controlling both cellular particulate organic and inorganic carbon content, whereas nutrient concentrations were the most important regulator of cellular particulate nitrogen and phosphorus of *E. huxleyi*. In contrast, elevated 30 $pCO_2$ had the greatest influence on cellular particulate inorganic carbon to organic carbon ratio,

resulting in a decrease in the ratio. Our results indicate that the different environmental drivers play specific roles in regulating the elemental composition of *E. huxleyi* with wide-reaching implications for coccolithophore-related marine biogeochemical cycles, as a consequence of the regulation of *E. huxleyi* physiological processes.

## 1 Introduction

The global climate change induced by anthropogenic activities is causing a wide range of alterations to the marine environment including ocean acidification (OA), rising sea surface temperature (SST), and intensified stratification due to increased density gradients between surface and subsurface waters, with associated shifts in mean irradiance levels and nutrient availability in the upper water column (Boyd and Doney, 2002; Rost and Riebesell, 2004; Stocker, 2013). All these global changes in environmental variables will affect the physiology and ecology of phytoplankton, both individually and interactively, in a complex way (Boyd and Hutchins, 2012; Boyd et al., 2010; Boyd et al., 2016; Feng et al., 2017).

Phytoplankton elemental composition is an important cellular property that reflects the metabolic rates of phytoplankton (Raven and Geider, 1988). Elemental composition is strongly influenced by environmental conditions and by phytoplankton adaptations to these conditions (Sterner and Elser, 2002) which in turn influences marine foodweb structure, particulate carbon export to the deep ocean, and ultimately marine biogeochemistry (Finkel et al., 2010 and references therein). The widely-recognized average molar elemental ratio of C:N:P is 106:16:1 for marine phytoplankton assemblages - the Redfield ratio (Redfield et al., 1963). However, individual phytoplankton species may have elemental ratios deviating, on short timescales (days to months) from Redfield depending on the environmental conditions they encounter. Such deviations subsequently influence the accumulation of these elements in the upper food web and also marine biogeochemistry (Finkel et al., 2010; Ho et al., 2003; Sardans et al., 2012).

Different environmental drivers may play a range of roles in regulating the stoichiometry of marine phytoplankton. Nutrient availability (Hecky et al., 1993; Perry, 1976) has been proven to affect phytoplankton stoichiometry directly. Irradiance provides the energy source for nutrient assimilation in

the cells (Goldman, 1986). In addition, temperature changes, which mainly alter metabolic rates, can also influence the diffusive uptake of nutrients into cells (Raven and Geider, 1988; Roleda et al., 2013). Increased levels of dissolved $CO_2$ during cell growth may result in higher cellular C:N and C:P ratios, due to increased $CO_2$ availability as a substrate for photosynthesis (Beardall et al., 2009; Feng et al., 2008; Fu et al., 2007; Fu et al., 2008). However, the dependency of C:N or C:P ratios on $CO_2$ availability can be species-specific (Burkhardt and Riebesell, 1997; Burkhardt et al., 1999). The effect of rising $p$CO$_2$ on the N:P ratio is still unclear, due to large variations observed in previous environmental manipulation studies (Sardans et al., 2012). For example, the N:P ratio of *Synechococcus* increased with elevated $CO_2$ concentration, but remained unchanged for *Prochlorococcus* (Fu et al., 2007) and *Emiliania huxleyi* (Feng et al., 2008).

Marine coccolithophores are responsible for almost half of the global marine calcium carbonate production, and are important in the marine carbon cycle through both the organic carbon pump and the inorganic carbon counter pump (Rost and Riebesell, 2004). *Emiliania huxleyi* is the most widely distributed coccolithophore species (Balch et al., 1991; Holligan et al., 1993; Holligan et al., 1983), and has been selected as a model phytoplankton species in the context of the marine carbon cycle (Westbroek et al., 1993). A wide range of environmental drivers, such as $CO_2$ concentration, nutrient level, irradiance, and temperature influence the growth, photosynthesis, and calcification of *E. huxleyi* both individually and interactively (Feng et al., 2017; Raven and Crawfurd, 2012; Zondervan, 2007). Changes in these physiological processes may in turn alter the elemental stoichiometry and composition of coccolithophores. Knowledge of how different environmental drivers will affect the elemental composition of *E. huxleyi* is important for a more complete understanding of the physiological responses of this species to the changing environment and the consequent effects on biogeochemical cycles. In addition, the magnitude of change in each environmental driver will be different with the future climate change, depending on location and scenario; hence a systematic study across a gradient of each driver is required.

This study advances previous findings by relating the change in elemental composition of *E. huxleyi* cells, in response to environmental forcing, to the physiological rate responses presented in the study of Feng et al. (2017). The major objective of the present study is to investigate and rank the importance of

the environmental drivers, including the nitrate and phosphate concentrations, irradiance, temperature, and $pCO_2$, on setting the elemental composition of a Southern Ocean strain of *E. huxleyi*. The combined results of this study and Feng et al. (2017) provide new insights into how environmental changes will impact the marine biogeochemical cycles related to *E. huxleyi*.

## 2 Materials and methods

### 2.1 Experimental setup

The marine coccolithophore *Emiliania huxleyi* (morphotype A, strain NIWA1108) was isolated from the surface water (depth of 5-6 m, salinity of 34.78) at 41°35.8'S 175°41.5'E, east of New Zealand by Dr. H. Chang aboard the *RV Tangaroa* on the research voyage TAN0909 in November 2009. The water
temperature was 12.1°C at the sampling site. The stock culture was maintained in the laboratory at 14°C and an irradiance of ~140 µmol m$^{-2}$ s$^{-1}$, under a light/dark cycle of 12 h/12 h. The medium used for maintaining the stock culture was seawater obtained from Otago Harbour, New Zealand (nutrient concentrations: phosphate 0.3-0.6 µM, nitrate 3-6 µM), filtered using 0.2 µm pore size filtration cartridge (Whatman$^{TM}$) and supplemented with nutrient stock solutions to give final concentrations of
nitrate 96 µM and phosphate 6 µM, without silicate addition. Trace metal and vitamin stock solutions were added according to the f/20 recipe for both stock culture and the manipulation experiments (10 times dilution of f/2 level; Guillard and Ryther, 1962).

For the manipulation experiments, *E. huxleyi* cells, in exponential growth phase determined by the growth curve, were transferred into acid-cleaned 500-mL polycarbonate bottles with screw caps and
20 subjected to a series of semi-continuous incubation experiments under different nutrient, irradiance, temperature, and $pCO_2$ conditions (Feng et al., 2017). Only one environmental driver was manipulated at a time for each incubation experiment, with the other environmental drivers remaining the same as the stock culture conditions. The manipulation of each of the different drivers was carefully selected to cover a broad range of conditions observed in the natural environment and those commonly employed
for laboratory incubations (Feng et al., 2017). Initial cell abundances were ~10$^4$ cell mL$^{-1}$ and *in vivo* chlorophyll *a* (Chl-*a*) fluorescence readings were monitored daily as indicators of Chl-*a* biomass and

cell growth. After 2-3 days of acclimation under the experimental conditions daily dilution was started by adding freshly made seawater medium into each incubation bottle to adjust the biomass to the level of the previous day. All the incubation experiments were carried out in walk-in growth chambers (Model 650, Contherm, New Zealand), with metal halide lamps (full spectrum) as the light source, under a light/dark cycle of 12 h/12 h. Irradiance levels inside the incubation bottles were measured using a quantum PAR sensor (2100 series, Biospherical Instruments Inc., USA). The temperature in each incubation experiment was monitored continuously using a HOBO Pendant[®] temperature/light data logger (Onset, Australia), with variation no more than ± 0.5°C.

The experimental conditions for each environmental driver used for the manipulation experiments are presented in Table 1, and described in Feng et al. (2017). All the treatments were conducted in triplicate. For each culture, the final sampling was performed after the daily monitored growth rate remained relatively constant (daily variations <10%) for more than seven generations (Feng et al., 2008). This yielded total acclimation of the cultures to the experimental conditions for ~20 days. Samples were collected for cell counts, Chl-*a* biomass, and elemental components, including particulate organic carbon (POC), particulate inorganic carbon (PIC), particulate organic nitrogen (PON), and particulate organic phosphorus (POP), starting 2 hours after the beginning of the light incubation phase and finishing within 2 hours for all the experimental treatments.

## 2.2 Sample analyses

### 2.2.1 Cell counts and Chl-*a*

One mL subsamples for cell counts were taken from each incubation bottle, preserved by adding 6 µL modified Lugol's solution, prepared by dissolving 10 g KI and 5 g iodine crystals in 20 mL Milli-Q water, then adding 50 mL of 5% anhydrous sodium acetate solution, and stored in dark at 4°C for no more than 5 days before counting. Cell abundance was determined with a nannoplankton counting chamber (PhycoTech, USA) using a Zeiss microscope (Axiostar plus, Germany). *In vitro* Chl-*a* concentration was analysed using a Turner 10-AU fluorometer (Turner design, USA) with 90% acetone extracted samples, as detailed in Welschmeyer (1994).

### 2.2.2 Elemental composition

Subsamples from each incubation bottle for PON, total particulate carbon (TPC), and POC measurement were filtered onto pre-combusted GF/F glass fibre filters (Whatman[TM]), and analysed using an elemental combustion system (Elementar vario EL III, Germany). Filters for POC analysis, were fumed with saturated HCl for 3 hours in order to remove all the inorganic carbon (Zondervan et al., 2002). The PIC content was calculated by subtracting POC from TPC values. Subsamples for particulate organic phosphate (POP) measurements were filtered onto pre-combusted GF/F filters (Whatman[TM]), and analysed following the molybdate colorimetric method of Solórzano and Sharp (1980). The particulate inorganic nitrogen (PIN) and particulate inorganic phosphorus (PIP) contents were both neglected due to their relatively low values for laboratory cultures (Feng et al., 2008).

### 2.2.3 Data analyses

The effects of different environmental drivers on the elemental composition of *E. huxleyi* and their stoichiometric ratios were identified with one-way analysis of variance (ANOVA) using the commercial statistical software package SigmaStat (Version 3.5; Jandel Scientific, San Rafael, CA, US). Differences between treatments were considered significant when $p < 0.05$. Post-hoc comparisons using the Student-Neuman-Keuls (SNK) test were conducted to determine any differences between particular treatments.

For the environmental drivers that had significant effects on the elemental composition and ratios of *E. huxleyi* within the examined range, the response curves to the drivers were fitted using the models listed in Table S1. All of the curve-fitting was performed using least square fit with Prism software (version 5.0; GraphPad Prism Software, US) with all the triplicate data for each of the experimental treatments.

The same approach as used in Feng et al. (2017) was performed to rank the relative importance of each environmental driver, that was found to have significant effects using the one-way ANOVA analyses, on the elemental composition of *E. huxleyi*. Firstly, the two values for the elemental composition at the average present-day conditions and the projected conditions for the year 2100 were derived from each fitted dose-response curve for the environmental driver that had significant effects.

The environmental conditions were projected using Coupled Model Intercomparison Project phase 5 (CMIP5) models (Boyd and Law, 2011; Law et al., 2016; Rickard et al., 2016), which suggested 33% decreases in both nitrate and phosphate concentrations, 2°C warming, a 25% increase in irradiance, and an increase in $p$CO$_2$ from 39 to 75 Pa in the Chatham Rise area for the year 2100 compared to present day conditions (Feng et al., 2017). The magnitude and direction of percentage change in the elemental composition under projected future conditions relative to the present day condition for each environmental driver was then calculated. The ranking was finally determined by comparing the absolute values of the calculated percentage changes of the physiological metrics caused by each driver. The driver that caused the largest percentage change was selected as the most important controlling driver.

## 3 Results

### 3.1 Changes in cellular POC content in response to environmental drivers

Cellular POC content was significantly affected by alteration of irradiance, temperature, and $p$CO$_2$ (Fig. 1). Increasing irradiance from 14 to 80 μmol photons m$^{-2}$ s$^{-1}$ increased the cellular POC content by around two-fold from $8.20 \pm 2.39$ to $14.07 \pm 1.17$ pg cell$^{-1}$ ($p<0.05$). POC content decreased at the two highest irradiance levels (350 and 650 μmol photons m$^{-2}$ s$^{-1}$, Fig. 1c). A trend of decreased $E.$ $huxleyi$ cellular POC content with elevated temperature was evident from the temperature manipulation experiment (Fig. 1d). The cellular POC content ($28.85 \pm 6.98$ pg cell$^{-1}$) was significantly higher than all the other treatments ($p<0.05$) at the lowest temperature of 4°C and significantly reduced by ~ 70% at both 20°C and 25°C ($p<0.05$). Raising $p$CO$_2$ from 8 to 15 Pa significantly increased the cellular POC content from $9.63 \pm 1.67$ to $12.93 \pm 1.84$ pg cell$^{-1}$ (Fig. 1e), with cellular POC content being relatively uniform from 15 to 109 Pa.

### 3.2 Alteration of cellular PIC content in response to environmental drivers

Temperature was the only driver that significantly altered the cellular PIC content (Fig. 2). There was a general trend of decreased cellular PIC content of $E.$ $huxleyi$ with warming from 11°C to 20°C (Fig. 2d). The cellular PIC content was significantly lower at 20°C and 25°C compared to the other four

temperature treatments ($p<0.05$). More than a 50% decrease in cellular PIC content was observed at the two highest temperature conditions, relative to the 7°C treatment. However, there were no significant differences in cellular PIC content between the other temperature treatments. The fitted $Q_c$ value (the plateau for one phase decay) was $6.94 \pm 0.93$ pg cell$^{-1}$, close to the average value at the two highest temperatures (Table S1).

**3.3 Changes in the cellular PIC:POC ratio in response to environmental drivers**

As for POC, the cellular ratio of PIC:POC was mainly affected by changes in irradiance, temperature, and $p$CO$_2$ (Fig. 3). The highest cellular PIC:POC ratio of $1.20 \pm 0.09$ was observed at the lowest irradiance (19 μmol photons m$^{-2}$ s$^{-1}$; $p<0.05$, compared to all other irradiance treatments). The ratio then decreased with increasing irradiance to $0.72 \pm 0.10$ at 190 μmol photons m$^{-2}$ s$^{-1}$ and slightly increased again at the two highest irradiances ($p<0.05$ between 190 and 650 μmol photons m$^{-2}$ s$^{-1}$, Fig. 3c). In the temperature manipulation experiment, the PIC:POC ratio was significantly lower ($p<0.05$) at the lowest temperature (4°C) than any other treatment, with a value of $0.45 \pm 0.03$ pg cell$^{-1}$. The PIC:POC value then leveled off between the range of 7°C to 25°C, with the average value more than double that at 4°C (Fig. 3d). With the variation of $p$CO$_2$ levels, the cellular PIC:POC ratio decreased by more than 40% from $1.46 \pm 0.02$ pg cell$^{-1}$ at 8 Pa to $0.90 \pm 0.15$ at 39 Pa and stayed similar between the range of 39 and 109 Pa ($p<0.05$) (Fig. 3e), mainly due to the increased cellular POC quota with rising $p$CO$_2$.

**3.4 Alteration of cellular PON content in response to environmental drivers**

The cellular PON content increased with increasing nitrate concentration. The content at the two lowest nitrate concentrations of 3.7 and 6.0 μM was less than half of the average value ($2.06 \pm 0.36$ pg cell$^{-1}$) of the three highest nitrate treatments (Fig. 4a). Warming from 4°C to 25°C decreased the cellular PON content ($p<0.05$). The value of $4.07 \pm 0.00$ pg cell$^{-1}$ at 4°C was double that at 15°C ($1.93 \pm 0.10$ pg cell$^{-1}$) and three-fold greater than the PON content of $1.31 \pm 0.24$ pg cell$^{-1}$ at 25°C (Fig. 4d).

**3.5 Changes in cellular POP content in response to environmental drivers**

The cellular POP content of *E. huxleyi* was significantly altered by nitrate, phosphate, temperature, and $pCO_2$. POP content was slightly less at the three low nitrate concentrations (3.7, 6.0, and 12 µM), compared to those at 96 and 200 µM (p<0.05; Fig. 5a). Cellular POP content significantly increased with rising phosphate concentration (Fig. 5b), with the highest POP content observed at 20 µM phosphate. As observed for cellular POC and cellular PON contents, warming greatly decreased the cellular POP content (Fig. 5d), with a reduction of 65% from $1.08 \pm 0.14$ pg cell$^{-1}$ at 4°C to $0.38 \pm 0.04$ pg cell$^{-1}$ at 11°C, but then only a further decrease of ~0.1 pg cell$^{-1}$ from 15°C to 25°C. Significant differences in POP content were detected between the two lowest temperature treatments compared to all others. Conversely, with rising $pCO_2$ level there was a trend of increased cellular POP content (Fig. 5e), which almost doubled from $0.20 \pm 0.04$ pg cell$^{-1}$ at 8 Pa to $0.38 \pm 0.02$ pg cell$^{-1}$ at 109 Pa (p<0.05).

**3.6 Alteration of cellular C to Chl-*a* ratio in response to environmental drivers**

Alteration of all the five environmental drivers greatly affected the cellular ratio of POC to Chl-*a* content (C:Chl-*a*, g:g) (p<0.05) (Fig. 6). C:Chl-*a* decreased exponentially with increased nitrate concentration up to 50 µM, but stabilised between 50 and 200 µM (Fig. 6a). The highest ratio of $422.36 \pm 74.28$ was observed at the lowest nitrate concentration of 3.7 µM, significantly higher than all other treatments (p<0.05). The ratio then decreased by 87% at 200 µM. An increase in phosphate concentration, however, only slightly decreased the C:Chl-*a* ratio (Fig. 6b). Compared to the ratios at the two lowest concentrations, a significant decrease (p<0.05) at 6.0 µM and 20 µM was observed (by ~20% each). Increased irradiance increased the C:Chl-*a* ratio linearly, with more than a doubling at 650 µmol photons m$^{-2}$ s$^{-1}$ compared to the ratio of $47.45 \pm 12.58$ at 14 µmol photons m$^{-2}$ s$^{-1}$ (Fig. 6c). The C:Chl-*a* ratio dramatically decreased with warming, especially between 4°C and 7°C (Fig. 6d). The ratio of $131.26 \pm 42.96$ observed at 4°C was significantly higher than all the other temperatures (p<0.05). Significantly lower C:Chl-*a* ratios were observed at the two lowest $pCO_2$ levels of 8 and 15 Pa compared with the other treatments (p<0.05, Fig. 6e), with the ratio increasing by 42% from low to high $pCO_2$.

**3.7 Shifts in cellular elemental molar ratios in response to environmental drivers**

The PON to POP (N:P) ratio was significantly lower ($p<0.05$) at the two lowest nitrate treatments compared to the others (Table 2). In contrast, the POC to PON (C:N) ratio was significantly higher ($p<0.05$) at the two lowest nitrate concentrations. There was no significant difference in C:N ratios across the other four nitrate treatments ($p>0.05$). Changes in nitrate concentration did not significantly affect the POC to POP (C:P) ratio.

The N:P ratio of *E. huxleyi* increased at low phosphate concentrations (0.4 and 2 µM), with highest value in the 0.4 µM phosphate treatment ($p<0.05$). There was a significant increase in the C:P ratio ($p<0.05$) at the two lowest phosphate concentrations compared to the others. The highest C:P ratio, recorded at the lowest phosphate concentration (0.4 µM), was almost double the value at 2 µM and more than three times the average ratio of the other treatments (Table 2). In contrast, there were no significant differences in the calculated C:N ratio across the phosphate treatments ($p>0.05$).

Decreased C:N ratios were observed for low irradiances; the value at 14 µmol photons $m^{-2}$ $s^{-1}$ being significantly lower than the three highest irradiances ($p<0.05$). Similarly, a decreased C:P ratio was found at low irradiance, with a significantly lower value at 14 µmol photons $m^{-2}$ $s^{-1}$ compared to the three highest irradiances. Warming significantly increased the N:P ratio from 4°C to 20°C (Table 2).

**3.8 Ranking the importance of environmental drivers in altering *Emiliania huxleyi* elemental composition**

Ranking the response of the Southern Ocean *E. huxleyi* isolated to projected future changes in oceanic properties revealed differential responses between drivers and processes (Table 3, Fig. 7). Cellular POC and cellular PIC:POC ratio were both significantly influenced by $CO_2$ and temperature, with temperature affecting cellular POC content the most, while $CO_2$ was the most important factor regulating PIC:POC. However, only one driver (temperature) significantly regulated cellular PIC, with a 4°C warming causing a 14.2 % decrease. The cellular PON content was significantly affected by future nitrate concentration and temperature, with nitrate ranking the most important. Four (phosphate, temperature, $CO_2$, and nitrate) out the five environmental drivers, under end of the century conditions, significantly affected cellular POP content, with future phosphate concentration playing the most

important role. The rankings associated with statistically non-significant differences among the treatment intervals, as marked in Table 3 and Fig. 7, need to be considered with caution (see Feng et al. (2017).

## 4 Discussion

This is the first detailed study of the individual effect of five environmental drivers (nitrate concentration, phosphate concentration, irradiance, temperature, and $p$CO$_2$) on the cellular elemental composition of the coccolithophore *E. huxleyi*. Moreover, it is the first to rank the importance of the predicted changes in these environmental drivers on *E. huxleyi* elemental stoichiometry for the year 2100 relative to the present-day conditions. Relating changes in elemental composition is an important
addition to the responses of growth, photosynthesis, and calcification rates (Feng et al., 2017), providing insights into the biogeochemical consequences of the physiological effects induced by change in the five essential environmental drivers.

### 4.1 Effects of nutrient concentration on the elemental stoichiometry of *Emiliania huxleyi*

    The PON and POP cell quotas of *E. huxleyi* in the present study were mainly controlled by nitrate
and phosphate concentrations, respectively, as phytoplankton relies on seawater nutrient availability as the external elemental source (Hecky et al., 1993; Price, 2005; Sakshaug and Holmhansen, 1977). Nitrate concentration plays an important role in regulating the growth, photosynthetic, and calcification rates of *E. huxleyi* (Feng et al., 2017); however, the three lowest nitrate concentrations only resulted in slightly decreased cellular POP contents and had no significant effect on cellular POC or PIC content.
This indicates that the regulation of the nitrate concentration on the POC and PIC productivity in our study was mainly a consequence of decreased growth rate of the cells under nitrate limitation, as shown by Feng et al. (2017). This finding is in contrast to Paasche (1998) who observed higher *E. huxleyi* cellular PIC:POC ratios under nitrate limitation as a result of decreased cellular POC and increased coccolith abundance per cell in *E. huxleyi* strain BOF 92 isolated from the North Atlantic. Higher
PIC:POC ratios under nitrate limitation was alternatively attributed to increased calcite mass per lith of *E. huxleyi* strain CCMP 378 isolated from the Gulf of Maine (Fritz, 1999). In addition, phosphate

concentration did not significantly affect *E. huxleyi* cellular carbon content nor the PIC:POC ratio of cells in the present study. However, Paasche (1998) observed greatly increased PIC content of *E. huxleyi* (strain BOF 92) under phosphate limiting conditions, and Riegman et al. (2000) observed that a greater increase PIC quotas under phosphate limitation than nitrate limitation for *E. huxleyi* (strain L).

These discrepancies between studies in the nitrate or phosphate effects on cellular PIC:POC ratio are mainly due to the different nutrient concentrations in the culturing media. Paasche (1998) observed an increase in *E. huxleyi* PIC cell quota under the stationary phase of batch incubation, i.e. when cell division ceased as nitrate dropped to ≤0.2 μM and phosphate dropped to ≤0.03 μM. This supports the findings of both Riegman et al. (2000) and Fritz (1999) who conducted continuous incubations with

high cell densities of *E. huxleyi*. These studies observed an increased cellular PIC content when phosphate concentration fell below 0.4 nM (Riegman et al., 2000) or nitrate concentration was below the detection limit (Fritz, 1999). However, the present study used a semi-continuous incubation method with higher and relatively steady nutrient concentrations (with lowest nitrate and phosphate concentrations of 3.6 and 0.4 μM, respectively) and the cells were grown and sampled at a healthy

exponential growth phase. Similarly, Müller et al. (2008) only found higher *E. huxleyi* (strain CCMP371) cellular calcite content during the stationary but not the exponential growth phase under both nitrate and phosphate limitation, due to the different cell cycle phases during which the calcification and cell division occurred. The authors explained that calcification continued during the G1 phase of cell assimilation when cell division was restricted under nutrient limitation, and thus the

cellular PIC content was increased (Müller et al., 2008). Further studies at extremely low nutrient concentrations (<0.1 μM) in a steady-state growth phase are still needed to understand the potential connection between carbon production and extreme nutrient limitation, given reports of areal expansion of oligotrophic waters in the world oceans with global climate change (Polovina et al., 2008).

**4.2 Irradiance effects on the elemental stoichiometry of *Emiliania huxleyi***

In the present study, irradiance was the main environmental factor affecting cellular POC content which in turn altered the PIC:POC ratio. The increased PIC:POC cellular ratio at low irradiance indicates that calcification is less dependent on irradiance than organic carbon fixation, as discussed in

Feng et al. (2017). Although both processes require light as an energy source, calcification requires less energy (Anning et al. 1996) than photosynthesis (Paasche, 1965; Balch et al., 1992). Therefore, the calcification rate is generally saturated at lower irradiance levels than photosynthesis (Paasche, 1964; Zondervan, 2007). Feng et al. (2017) reported greatly reduced photosynthetic rates under the two lowest irradiance levels, while observing that this trend was less significant for the calcification rate. Hence, limiting irradiance will lead to less POC content in the cells compared to the cellular PIC quota, and thus a higher cellular PIC:POC ratio would be expected at low irradiance when growth and photosynthesis are light-limited (Raven and Crawfurd, 2012), as confirmed by the response of calcification:photosynthesis in Feng et al. (2017).

Increasing irradiance also elevated the C:Chl-*a* ratio linearly in the present study, due to the increase in POC quota and a decrease in Chl-*a* quota, as also reported for diatoms and dinoflagellates (Geider, 1987). The reduced cellular pigment quota under high irradiance helps to reduce the energy required for light harvesting in phytoplankton cells, which is a strategy to balance the energy demands for growth and POC production with photon harvesting (Kiefer, 1993). In addition, the present study revealed that the C:N and C:P ratios of *E. huxleyi* both increased at high light levels, as a consequence of increased cellular POC content driven by increased irradiance but no significant change in cellular PON or POP quota, further suggesting that organic carbon content is more light dependent than the accumulation of cellular N or P (Geider et al., 1998).

**4.3 Temperature effects on the elemental stoichiometry of *Emiliania huxleyi***

Temperature is important in regulating dissolved chemical diffusion and transport, non-enzymatic and enzymatic reactions, and the metabolic rates of phytoplankton (Raven and Geider, 1988). In our accompanying study, the growth, photosynthetic, and calcification rates all increased with rising temperature until the optimal temperature was reached at 25°C, 24°C, and 20°C respectively (Feng et al., 2017), which were all higher than the stock culture growth temperature or the temperature at the isolation site of *E. huxleyi* strain NIWA 1108. In the present study, the cellular POC, PON, and POP content all reduced significantly as temperature increased. It has been proposed that reduced cell size is a universal strategy in response to increasing temperature for both terrestrial and aquatic organisms

(Gardner et al., 2011), following a hypothesis suggested by Atkinson et al. (2003). A study on the coccolithophores *E. huxleyi* (strain EH2) and *Gephyrocapsa oceanic* (strain GO1) observed decreased cell size and thinner coccospheres upon raising temperature from 10°C to 25°C, which was attributed to the relatively suppressed cell division at low temperature (Sorrosa et al., 2005). This decrease in cell
volume (Fig. S1) could be the main cause of reduced cellular elemental components in the present study. Previous studies also reported that warming resulted in reduced cell volume of *E. huxleyi* (strain AC481, De Bodt et al., 2010; strain L, van Rijssel and Gieskes, 2002), and decreased cellular POC and PIC quotas of coccolithophore *Coccolithus pelagicus* when the temperature was raised from 10°C to 15°C (Gerecht et al., 2014). Similarly, warming significantly decreased the cellular elemental contents
to their lowest levels measured in the present study over the range from 4°C to 25°C, with a decrease in cell size at higher temperatures (Fig. S1), as growth rate increased (Feng et al, 2017).

However, contrary to the observed changes in POC, PON, and POP cell quota, the cellular PIC content of *E. huxleyi* only decreased when temperature was higher than 11°C in the present study, due to the strongly reduced calcification and malformation at low temperatures of 4°C and 7°C (Feng et al.,
2017). The reduced cell division rate (i.e. enlarged cell volume, Fig. S1) offset the reduced calcification rate at lower temperatures, and so there was no significant difference in PIC cell quota at temperatures below 11°C. Consequently, the cellular PIC:POC ratio was lower at 4°C and 7°C, consistent with the trend observed for the calcification: photosynthesis ratio (Feng et al., 2017), indicating suppression of PIC formation relative to POC production at low temperature (Watabe and Wilbur, 1966). The PIC:POC
ratio then decreased with warming from 11°C to 15°C and remained relatively steady afterwards, mainly due to the lower optimal temperature for calcification (20°C) compared to photosynthesis (24°C) as suggested in Feng et al. (2017).

Furthermore, warming from 4°C to 20°C significantly increased the *E. huxleyi* cellular N:P ratio in the present study, in agreement with the recent model study on a natural phytoplankton community
(Toseland et al., 2013). Toseland et al. (2013) found that with increasing temperature the rate of cellular protein synthesis in phytoplankton was higher, but with a lower number of phosphorus-rich ribosomes, thereby increasing the cellular N:P ratio. In the present study, the cellular N:P ratio of *E. huxleyi* at 20°C increased by 74% from that at 4°C, in spite of both cellular PON and cellular POP decreasing with

warming. Although this study presents results for a single strain of *E. huxleyi*, if the temperature dependency of cellular resource allocation is a universal trend for all the *E. huxleyi* genotypes, we can speculate that the diverse *E. huxleyi* strains growing in different temperature regions might have different requirements for nitrogen *vs*. phosphorus, and that the growth of *E. huxleyi* strains in the temperate to tropical regions might be more readily limited by nitrate than sub-polar strains. Similarly, Toseland et al. (2013) suggested that future warming might accentuate nitrate limitation in the oceans.

## 4.4 Effects of $CO_2$ on the elemental stoichiometry of *Emiliania huxleyi*

The photosynthesis of *E. huxleyi* was saturated at a higher $pCO_2$ than that for growth rate (Feng et al., 2017). In the present study, $CO_2$ plays the most important role in regulating the cellular PIC:POC ratio. The PIC:POC ratio was significantly higher at the lowest $pCO_2$ level, as a consequence of the lower cellular POC and higher cellular PIC at 8 Pa. In general, cell growth of *E. huxleyi* is less limited by low $CO_2$ concentrations than in other phytoplankton groups (Clark and Flynn, 2000; Paasche et al., 1996; Riebesell et al., 2000a; Rost et al., 2003). Moreover, recent studies suggest that *E. huxleyi* operates an active carbon concentrating mechanism (CCM) to utilize $HCO_3^-$ through the enzyme carbonic anhydrase (CA; Reinfelder, 2011), and may have high affinity for $CO_2$ in photosynthesis (Stojkovic et al., 2013). However, the efficiency of CCMs in *E. huxleyi* (strain B92/11) is considered to be low as a consequence of the leakage of $CO_2$ from the cell (Rost et al., 2006), and so coccolithophore photosynthesis is more dependent than cell growth on $CO_2$ concentration (Rost and Riebesell, 2004). This discrepancy between growth and organic carbon fixation can lead to a decrease in cellular POC at low $pCO_2$. This difference in $CO_2$ requirements between the two processes may also have resulted in the lower cellular POP content at 8 Pa compared to other $pCO_2$ treatments.

The increasing trend observed for cellular POC and POP was not apparent for cellular PIC quota, as calcification rates significantly decreased with increasing $pCO_2$ level >40 Pa (Feng et al., 2017). Hence the cellular PIC:POC ratio was significantly higher at the two lowest $pCO_2$ levels, consistent with previous findings for $CO_2$ manipulations at saturating irradiances on *E. huxleyi* (strain PML B92/11A; Zondervan et al., 2002; Zondervan et al., 2001). No further significant change in cellular carbon content or PIC:POC ratio occurred at higher $pCO_2$, in contrast to the linear decrease in the

calcification:photosynthesis ratio with rising $p$CO$_2$ (Feng et al., 2017). This difference is noteworthy as both cellular PIC:POC and calcification:photosynthesis ratios are commonly used to examine the relative change of PIC and POC production in coccolithophores (Raven and Crawfurd, 2012). These changes have biogeochemical implications for the marine rain ratio in the carbon cycle (Klaas and Archer, 2002; Rost and Riebesell, 2004), which is the export ratio of calcite to organic carbon into the deep ocean. The [14]C-labelling technique used in this study (see Feng et al., 2017) to measure carbon fixation (photosynthesis and calcification) rates was conducted during the light period, thus the measured rate is an indicator of net carbon fixation, that does not account for the energy-consuming respiratory process or CO$_2$ leakage out of the cells (Bach et al., 2015, 2013; Rost et al., 2006). Conversely, the cellular carbon content indicates the gross accumulated carbon in the cells over longer period of growth (Engel et al., 2010; Fabry and Balch, 2010). The most compelling reason for the relatively higher PIC:POC ratio (~1.5) than calcification: photosynthesis ratio (~1.0) in the lowest pCO$_2$ treatment (8 Pa, 79 ppm) in our study may then be attributed to diffusive CO$_2$ loss limiting inorganic carbon active uptake from the substrate (Bach et al., 2013), resulting in less POC fixation into the cells relative to the PIC fixation by the calcification process.

The C:Chl-$a$ ratio of $E.$ $huxleyi$ was lowest at $p$CO$_2$ of 8 Pa across all the $p$CO$_2$ treatments in the present study, mainly due to the decreased cellular POC at low $p$CO$_2$, rather than any change in cellular Chl-$a$ content. However, increasing $p$CO$_2$ did not have significant effects on the C:N, N:P or C:P ratios in the present study. This is in accordance with a recent study on $E.$ $huxleyi$ (strain PML B92/11A), which also exhibited constant C:N:P ratios across a $p$CO$_2$ range of 18 to 75 Pa for cultures at steady growth phase under phosphate-limited continuous incubation (Engel et al., 2014).

**4.5 Biogeochemical implications and future directions**

The comparisons between present day conditions and those projected for year 2100 for the Chatham Rise area are summarized in the conceptual figure (Fig. 7). These results indicate that the 2°C warming will decrease both POC and PIC cellular quotas of $E.$ $huxleyi,$ but may slightly increase the PIC:POC ratio. Rising $p$CO$_2$ alone will result in decreased cellular PIC:POC ratio. Although the 33% decrease in nitrate concentration is the major factor controlling the growth, photosynthetic, and calcification rates

(Feng et al., 2017), change in nitrate concentration did not significantly affect the elemental stoichiometry except for the cellular PON contents of *E. huxleyi*. In addition, increasing temperature may increase the cellular N:P ratio, while rising $p$CO$_2$ will decrease the N:P and C:P ratios. These results provide a more detailed perspective that can improve our knowledge on how the model coccolithophore species, *E. huxleyi*, may respond to future environmental changes. For example, our results suggest that rising $p$CO$_2$ in the future oceanic environment will decrease the *E. huxleyi* cellular PIC:POC ratio by 5.4%; however, the projected warming and increase in irradiance level may offset this decreased PIC:POC by 2.4% and 0.3%, respectively. The changes in PIC:POC have implications for the marine "rain ratio" and so alter the marine carbon cycle (Rost and Riebesell, 2004). Similarly, the cellular N:P ratio will be decreased by rising $p$CO$_2$, although this trend may be canceled out by warming. The altered C:N:P stoichiometry will in turn affect the nutrient cycle at higher trophic levels (Jones and Flynn, 2005) and marine biogeochemical cycles (Beardall and Raven, 2004).

It is noteworthy that the research presented here only examined the physiological response norms of *E. huxleyi* to a single environmental driver when other drivers were all kept at the stock culture growth condition (i.e. a set of single dimensional space experiments). However, these responses (such as the shape of the curves and the optimal conditions) may be different when the other background conditions are changed. For example, Sett et al. (2014) observed the dose-response curves of calcification of *E. huxleyi* PML B9/11 to CO$_2$ concentration was regulated by temperature. Therefore, in order to comprehensively understand how *E. huxleyi* physiology will respond to multiple environmental drivers and fill this knowledge gap, future research on a full environmental matrix is still necessary. These experiments will not only help to further explore the potential interactions (i.e. synergistic or agnostic effects) between environmental drivers, but also provide a better understanding of the underlying mechanisms of these interactive effects. In addition, the present study is only based on a single strain of southern hemisphere *E. huxleyi*. Due to the wide distribution of this species in the natural marine environment, *E. huxleyi* presents high variability in terms of genetic, morphological, and physiological characteristics (Cook et al., 2011; Read et al., 2013; Young et al., 2014). Therefore, the physiology of different *E. huxleyi* strains isolated from different geographic locations might respond differently to changing environmental drivers. For example, within the context of OA research, extensive previous

studies suggest a strain-specificity of *E. huxleyi* in response to changes in seawater carbonate chemistry (Langer et al., 2009; Raven and Crawfurd, 2012; Blanco-Ameijerias et al., 2016). It has also been observed that different *E. huxleyi* ecotypes/morphotypes responded differently to OA (Müller et al., 2015), which is likely a consequence of their genetic variation (Cook et al., 2011). The present study and Feng et al. (2017) demonstrate the important roles of different environmental drivers in controlling the physiology of *E. huxleyi* strain NIWA1108, and so further work is required to determine if the findings apply to other strains.

In summary, this study, in combination with Feng et al. (2017), have a number of implications for research into the response of *E. huxleyi* to ocean acidification and global climate change. In addition to seawater carbonate chemistry (Riebesell et al., 2010), it is necessary to report the experimental conditions of all the environmental drivers carefully. The predictions presented will provide useful information for biogeochemical models, such as that of Bopp et al. (2001), of how the elemental stoichiometry of *E. huxleyi* will respond to the alteration of these environmental conditions individually, in order to predict the future changes in the marine biogeochemical cycles. In addition, multiple environmental drivers tend to change simultaneously in the future global climate change scenario (Boyd and Hutchins, 2012), and so future studies should also investigate the interactions between these multiple drivers on phytoplankton physiology. The predicted future changes in marine physical properties (such as sea surface temperature (SST) and mixed layer depth) will vary from one oceanic region to another (Boyd and Doney, 2002). The dose response curves from our study suggest that the range of alteration in environmental drivers may control the outcome of the effects of environmental perturbation on *E. huxleyi* physiology and biogeochemistry. For future multi-factorial manipulation experimental designs, our results suggest that the magnitudes of change in each environmental driver need to be determined/decided cautiously and should have environmental relevance in order to make more accurate predictions, and the understanding of interactive effects of multiple environmental drivers and the underlying mechanisms should be further explored.

## 5 Acknowledgements

We would like to thank Hoe Chang at National Institute of Water and Atmospheric Research Ltd. (NIWA) Wellington for providing the stock culture of *Emiliania huxleyi* NIWA1108. We also thank Kim Currie at NIWA, Department of Chemistry, University of Otago for helping with the DIC and alkalinity analysis and Graham Rickard for generating the future estimates of environmental variables for New Zealand waters from the CMIP5 models. This work was supported by New Zealand Marsden grant (09-UOO-175) to CLH and National Natural Science Foundation of China (No. 41306118 and No. 41676160) to YF.

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

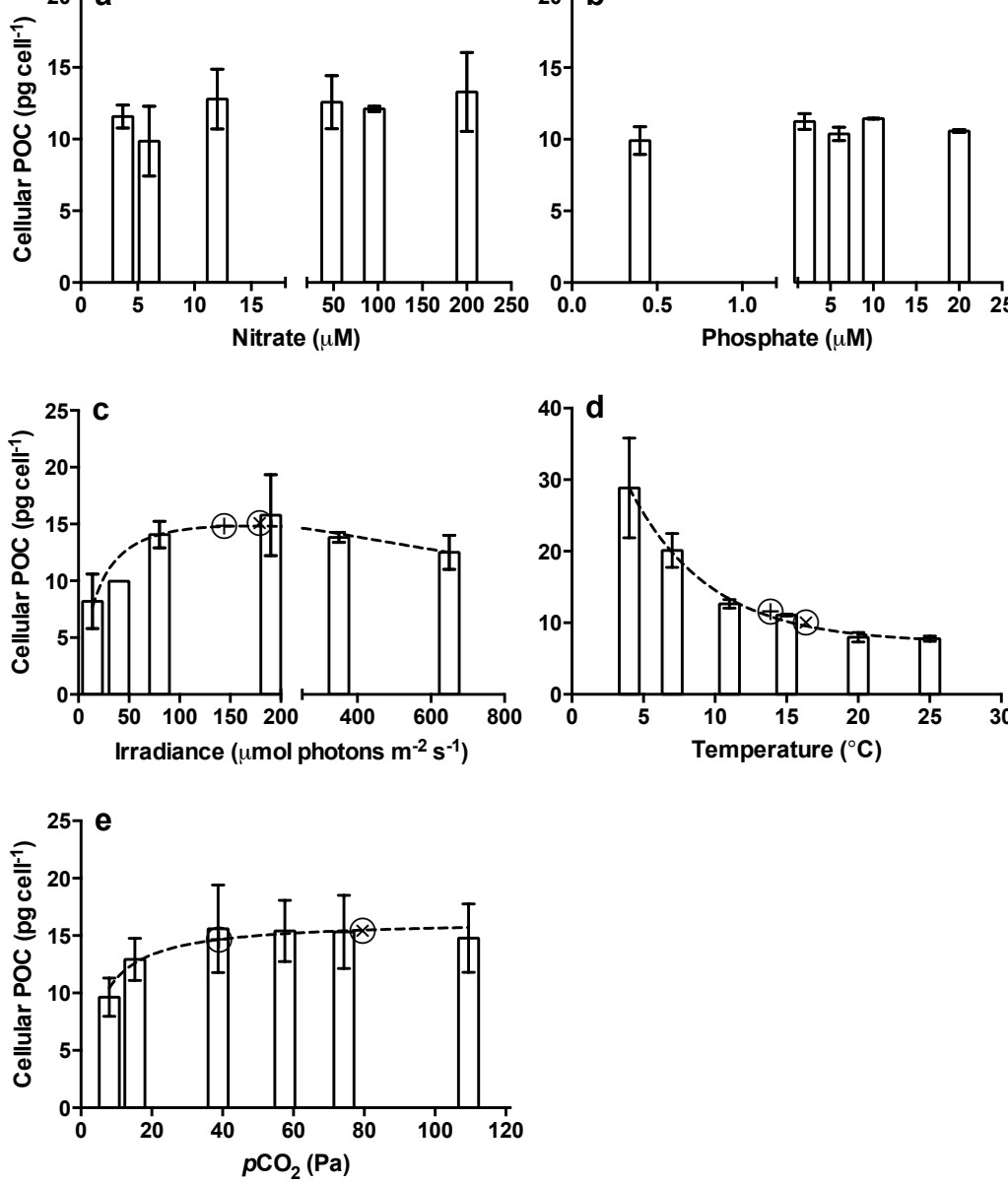

**Figure 1. Changes in *Emiliania huxleyi* cellular particulate organic carbon (POC) content in response to different environmental drivers: a) cellular POC *vs.* nitrate concentration; b) cellular POC *vs.* phosphate concentration; c) cellular POC *vs.* irradiance; d) cellular POC *vs.* temperature; and e) cellular POC *vs.* $pCO_2$. Error bars represent standard deviations (n=3).**

5   **For Figures1-5: The dashed lines represent the fitted dose-response curves. "⊕" represent the fitted values for the present day conditions in the Chatham Rise area, and "X" represent the fitted values for the predicted future conditions (2100) in the Chatham Rise area. Error bars represent standard deviation (n=3).**

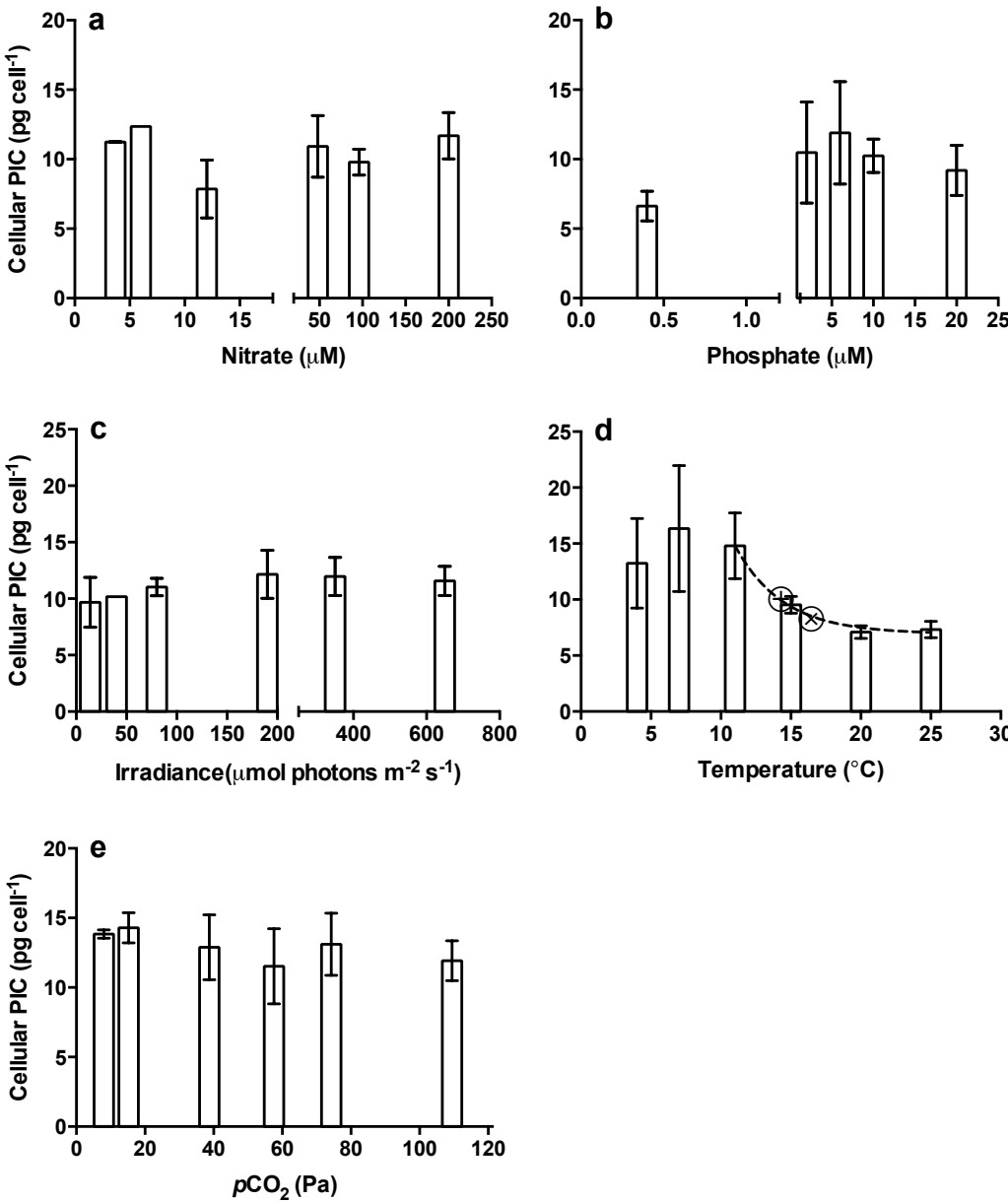

**Figure 2. Changes in *Emiliania huxleyi* cellular particulate inorganic carbon (PIC) content in response to different environmental drivers: a) cellular PIC *vs.* nitrate concentration; b) cellular PIC *vs.* phosphate concentration; c) cellular PIC *vs.* irradiance; d) cellular PIC *vs.* temperature; and e) cellular PIC *vs.* $pCO_2$.**

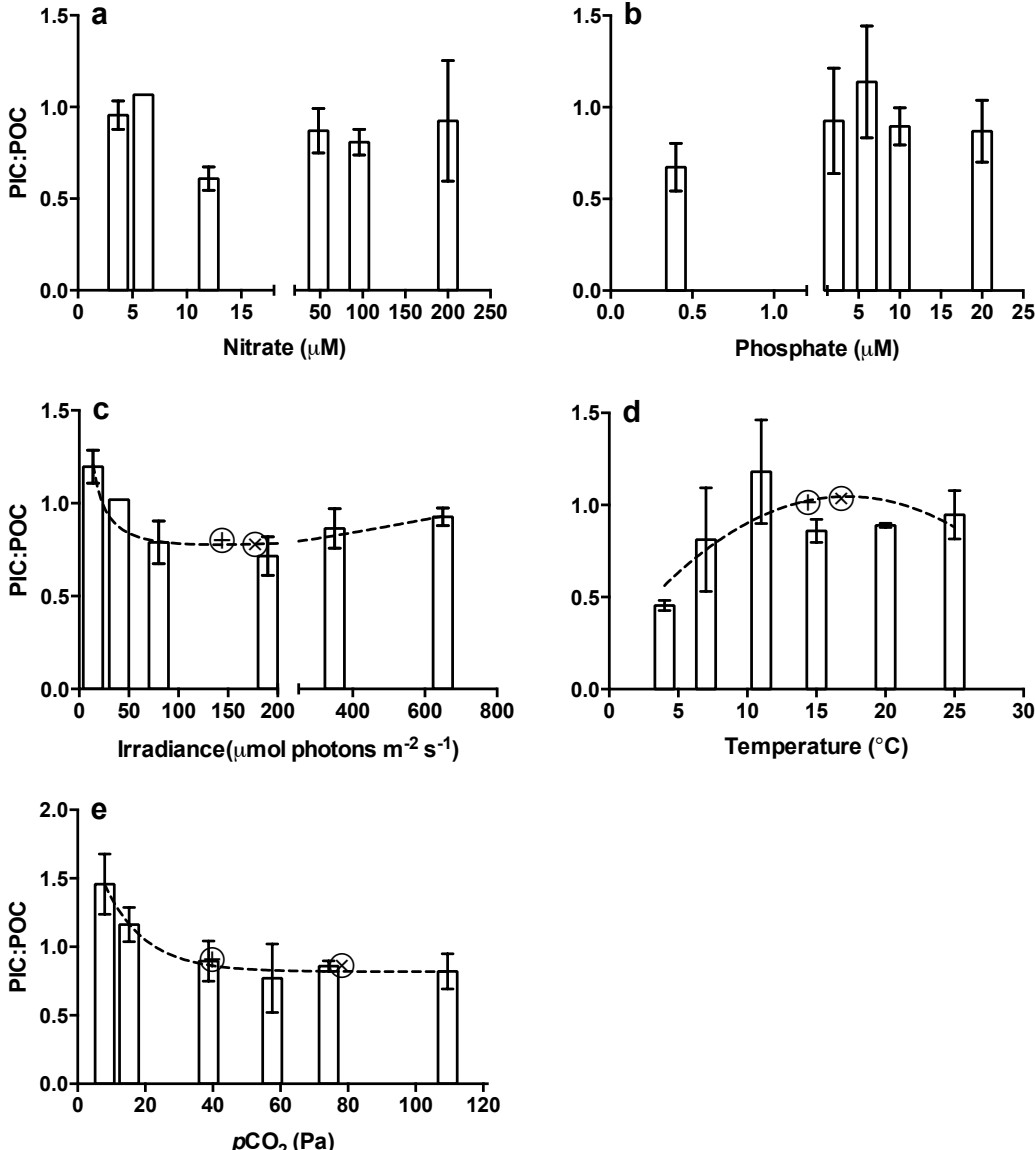

**Figure 3. Changes in the ratio of *Emiliania huxleyi* cellular particulate inorganic carbon content to particulate organic carbon content (PIC:POC) in response to different environmental drivers: a) PIC:POC ratio *vs.* nitrate concentration; b) PIC:POC ratio *vs.* phosphate concentration; c) PIC:POC ratio *vs.* irradiance; d) PIC:POC ratio *vs.* temperature; and e) PIC:POC ratio *vs.* $pCO_2$.**

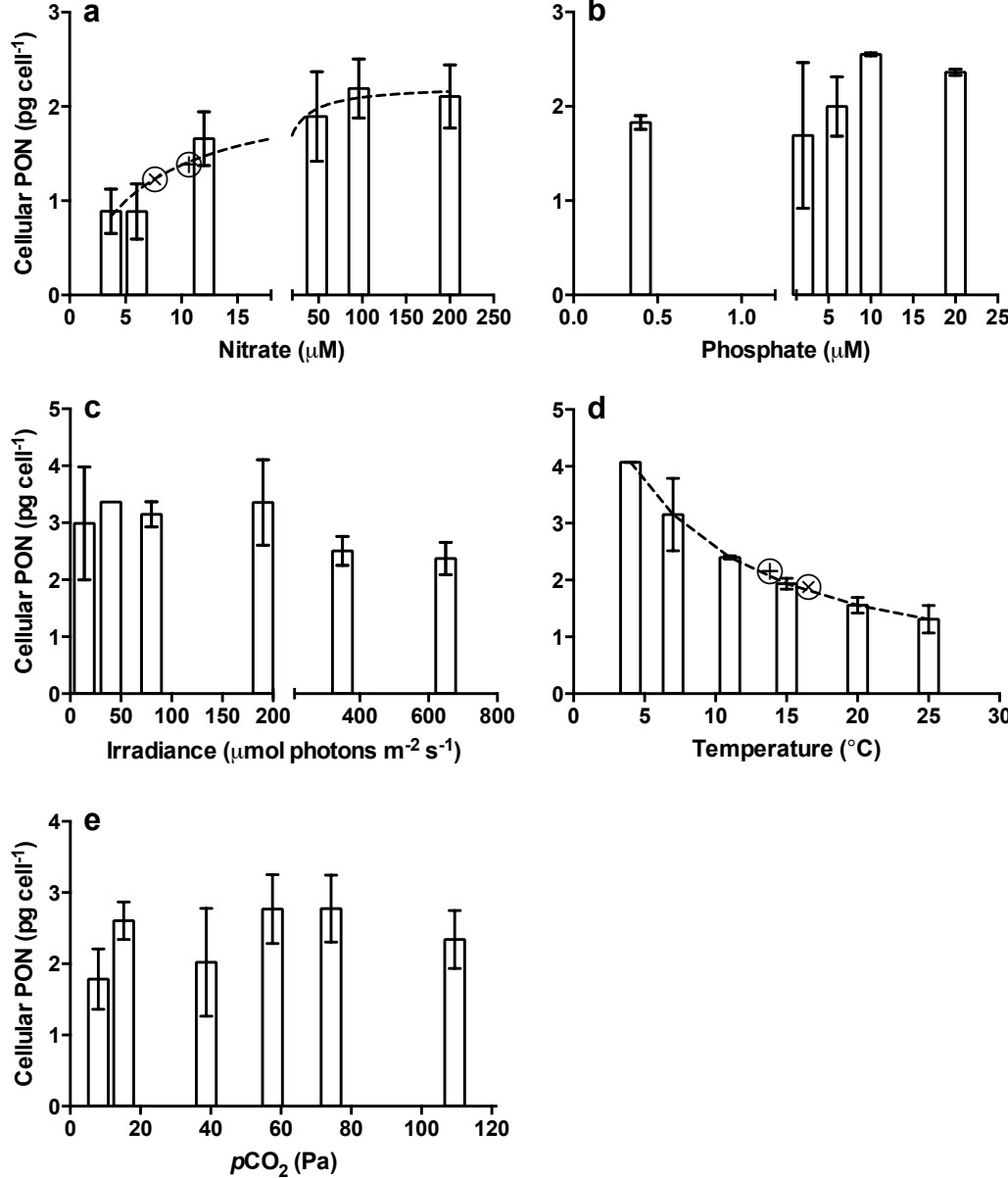

**Figure 4. Changes in *Emiliania huxleyi* cellular particulate organic nitrogen (PON) content in response to different environmental drivers: a) cellular PON *vs.* nitrate concentration; b) cellular PON *vs.* phosphate concentration; c) cellular PON *vs.* irradiance; d) cellular PON *vs.* temperature; and e) cellular PON *vs.* $pCO_2$.**

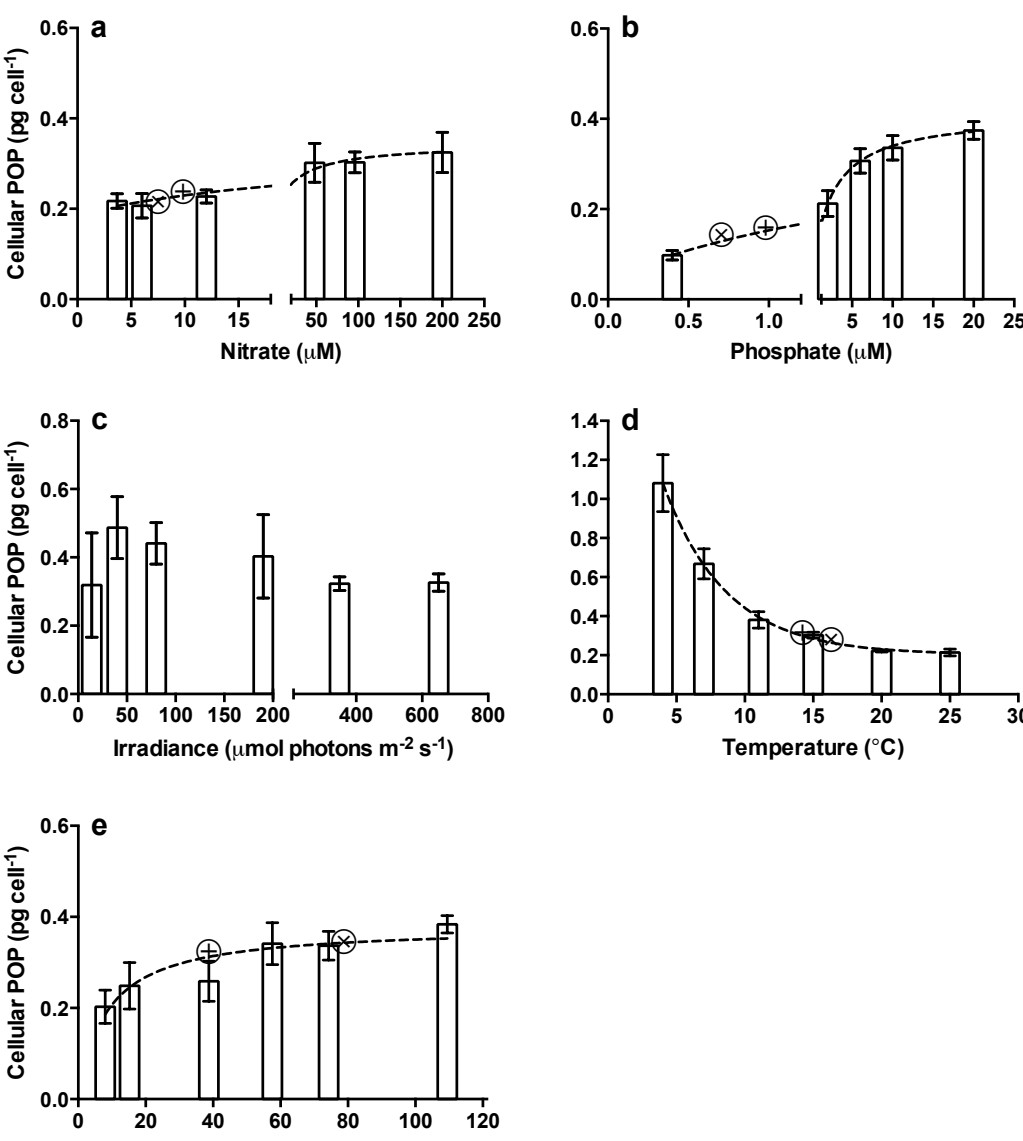

**Figure 5. Changes in *Emiliania huxleyi* cellular particulate organic phosphorus (POP) content in response to different environmental drivers: a) cellular POP *vs.* nitrate concentration; b) cellular POP *vs.* phosphate concentration; c) cellular POP *vs.* irradiance; d) cellular POP *vs.* temperature; and e) cellular POP *vs.* $p$CO$_2$.**

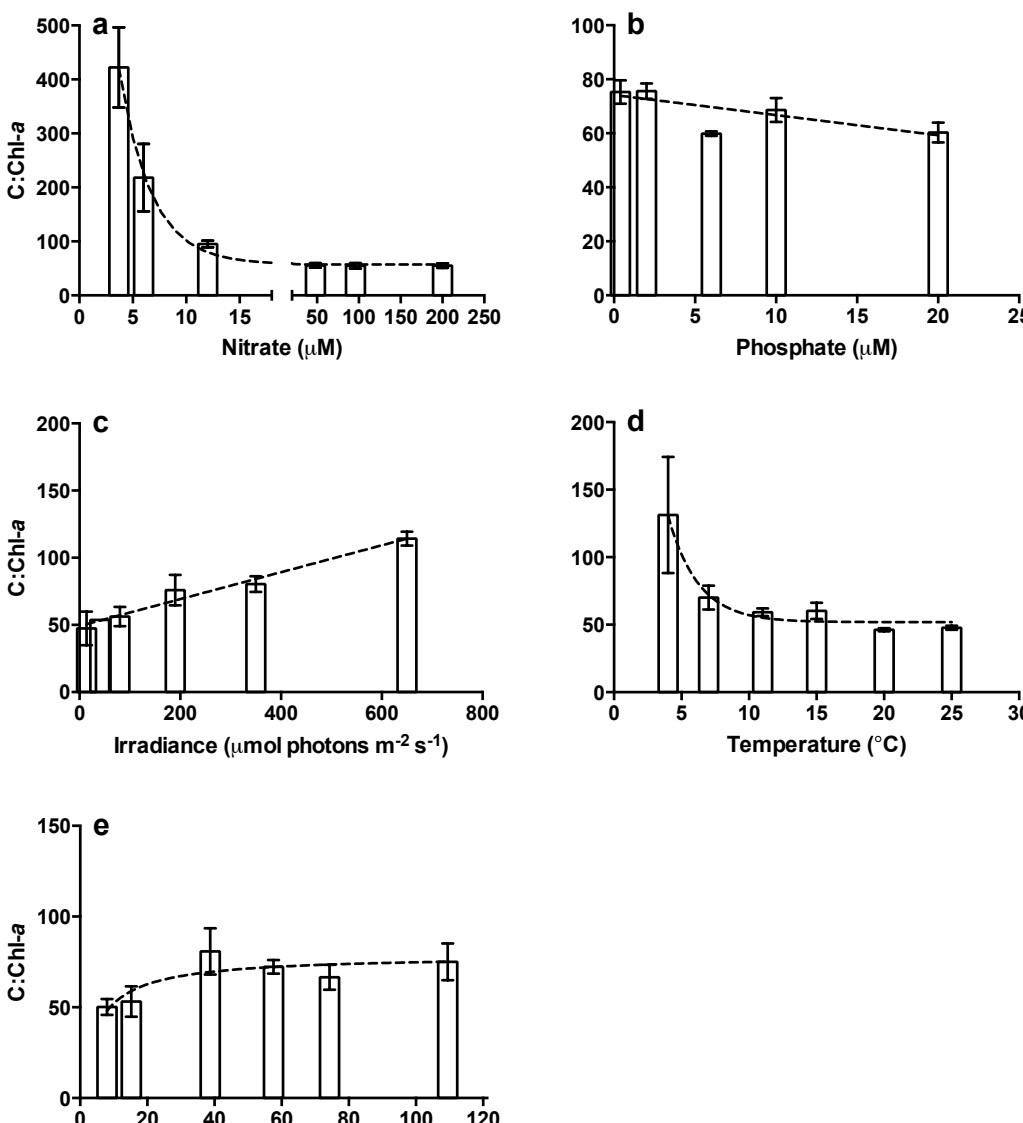

**Figure 6. Changes in the ratio of *Emiliania huxleyi* cellular particulate organic carbon content to chlorophyll *a* content (C:Chl-*a*) in response to different environmental drivers: a) C:Chl-*a* ratio *vs.* nitrate concentration; b) C:Chl-*a* ratio *vs.* phosphate concentration; c) C:Chl-*a* ratio *vs.* irradiance; d) C:Chl-*a* ratio *vs.* temperature; and e) C:Chl-*a* ratio *vs.* $p$CO_2. Error bars represent standard deviations (n=3).**

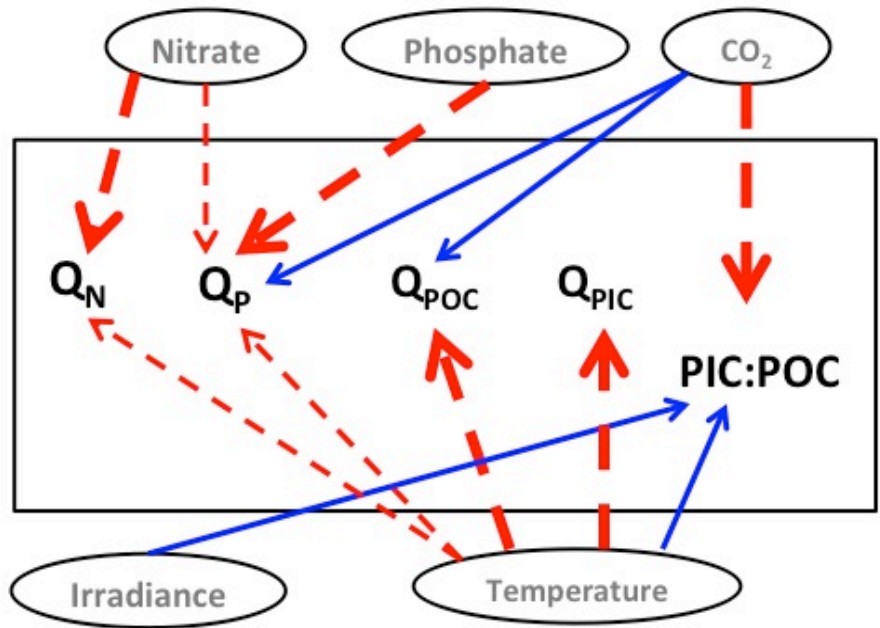

**Figure 7. Conceptual figure of the specific effects of each the five environmental drivers, under the projected future conditions (year 2100), on the elemental composition of *Emiliania huxleyi*. Q represents the cellular quota of each element of *Emiliania huxleyi*.**

5     **\*\*\*The box denotes the *E. huxleyi* cell. Solid blue arrows indicate positive effects of the future environmental changes; and dash red arrows indicate negative effects of the future environmental changes. Arrows in bold indicate the environmental drivers that play the most important role regulating the connected physiological metrics under the predicted environmental conditions for the year 2100.**

**Table 1. Treatment conditions for each environmental driver used in the manipulation experiments.**

| Environmental driver | Treatments |
|---|---|
| Nitrate ($\mu$M) | 3.7; 6; 12; 48; 96; 200 |
| Phosphate ($\mu$M) | 0.4; 2; 6; 10; 20 |
| Irradiance ($\mu$mol photons m$^{-2}$ s$^{-1}$) | 14; 40; 80; 190; 350; 650 |
| Temperature (°C) | 4; 7; 11; 15; 20; 25 |
| $p$CO$_2$ (Pa) | 8; 15; 39; 58; 74; 109 |

**Table 2.** Elemental molar ratios of N:P, C:N and C:P of *Emiliania huxleyi* from the five single-factorial manipulation experiments. The errors are standard deviations around the mean (n=3). The values in bold are significantly different compared to other treatments.

| Environmental driver | Treatment | N:P (mol:mol) | C:N (mol:mol) | C:P (mol:mol) |
|---|---|---|---|---|
| Nitrate (μM) | 3.7 | **9.09±2.48** | **15.90±4.09** | 137.69±8.27 |
| | 6 | **9.07±1.55** | **13.16±1.13** | 118.45±10.06 |
| | 12 | 15.90±1.38 | 9.00±0.07 | 143.08±11.28 |
| | 48 | 14.46±1.39 | 7.01±0.64 | 107.76±11.93 |
| | 96 | 16.16±3.45 | 6.56±1.07 | 103.56±6.54 |
| | 200 | 14.31±0.40 | 7.32±0.47 | 104.82±8.34 |
| Phosphate (μM) | 0.4 | **41.57±4.14** | 6.32±0.42 | **261.67±17.30** |
| | 2 | **21.47±0.28** | 9.27±5.08 | **137.57±11.31** |
| | 6 | 14.17±3.59 | 6.16±1.24 | 85.00±4.51 |
| | 10 | 17.06±2.06 | 5.24±0.06 | 89.28±9.80 |
| | 20 | 13.99±0.89 | 5.22±0.08 | 73.04±3.48 |
| Irradiance (μmol photons m$^{-2}$ s$^{-1}$) | 14 | 22.03±7.27 | **4.26±0.29** | 73.82±6.48 |
| | 40 | 14.27** | 4.38** | 62.50** |
| | 80 | 16.03±2.90 | 5.24±0.69 | 83.96±17.05 |
| | 190 | 18.71±1.74 | 5.99±0.23 | 112.39±14.60 |
| | 350 | 17.15±0.83 | 6.47±0.54 | 110.63±3.98 |
| | 650 | 16.11±1.77 | 5.70±0.31 | 91.49±6.66 |
| Temperature (°C) | 4 | 8.92±1.29 | 8.67±2.64 | 71.34±27.92 |
| | 7 | 10.46±2.05 | 7.60±1.32 | 78.70±14.78 |
| | 11 | 13.58±1.91 | 6.21±0.32 | 86.56±11.92 |
| | 15 | 14.12±0.66 | 6.70±0.31 | 94.47±3.84 |
| | 20 | 15.53±1.06 | 5.98±0.15 | 92.88±6.33 |
| | 25 | 13.67±2.99 | 7.08±1.39 | 93.96±4.87 |
| $p$CO$_2$ (Pa) | 8 | 19.39±2.41 | 6.81±1.09 | 122.61±9.97 |
| | 15 | 24.01±6.80 | 5.80±0.71 | 137.50±33.51 |
| | 39 | 16.96±3.62 | 7.41±0.06 | 155.64±31.42 |
| | 58 | 17.89±0.80 | 6.55±1.16 | 116.67±16.15 |
| | 74 | 18.22±2.45 | 6.42±0.43 | 116.93±17.60 |
| | 109 | 13.56±2.78 | 7.41±1.20 | 99.25±17.70 |

** Sample loss during analysis resulted in only single values at this irradiance.

**Table 3. Comparison of cellular particulate organic carbon (POC) contents, particulate inorganic carbon (PIC) contents, PIC:POC ratios, particulate organic nitrogen (PON) contents and particulate organic phosphorus (POP) contents of *Emiliania huxleyi* between projected (year 2100) and present day Chatham Rise conditions, with rankings of the importance of the environmental drivers which caused significant effects on each physiological parameter. The numbers of the ranking scheme represent the gradient of the most (1) to least (4) important effects. Effect of "+" represents an increase and "-" represents a decrease in the elemental composition/ratio in the future, respectively.**

| Physiological parameter | Environmental driver | Fitted values at different conditions of environmental drivers* | | Future vs. present day comparisons | | |
| --- | --- | --- | --- | --- | --- | --- |
| | | Present day | Future | Change (%)** | Effects (+/-) | Ranking |
| Cellular POC content (pg cell$^{-1}$) | Temperature | 10.798 | 9.713 | 10.0 | - | **1**[*] |
| | $CO_2$ | 14.632 | 15.436 | 5.5 | + | 2 |
| | Irradiance | 14.774 | 14.827 | 0.3 | + | 3 |
| | Nitrate | | | | | n.s. |
| | Phosphate | | | | | n.s. |
| Cellular PIC content (pg cell$^{-1}$) | Temperature | 10.206 | 8.753 | 14.2 | - | **1** |
| | Nitrate | | | | | n.s. |
| | Phosphate | | | | | n.s. |
| | Irradiance | | | | | n.s. |
| | $CO_2$ | | | | | n.s. |
| PIC:POC | $CO_2$ | 0.868 | 0.821 | 5.4 | - | 1 |
| | Temperature | 1.017 | 1.042 | 2.4 | + | 2 |
| | Irradiance | 0.777 | 0.780 | 0.3 | + | 3 |
| | Nitrate | | | | | n.s. |
| | Phosphate | | | | | n.s. |
| Cellular PON content (pg cell$^{-1}$) | Nitrate | 1.380 | 1.162 | 15.8 | - | **1** |
| | Temperature | 2.013 | 1.819 | 9.6 | - | **2** |
| | Phosphate | | | | | n.s. |
| | Irradiance | | | | | n.s. |
| | $CO_2$ | | | | | n.s. |
| Cellular POP content (pg cell$^{-1}$) | Phosphate | 0.106 | 0.078 | 25.9 | - | **1** |
| | Temperature | 0.304 | 0.269 | 11.6 | - | **2** |
| | $CO_2$ | 0.312 | 0.342 | 9.6 | + | **3** |
| | Nitrate | 0.249 | 0.227 | 8.9 | - | 4 |
| | Irradiance | | | | | n.s. |

*The fitted values for "present day" and "future" were extracted from the fitted dose-response curves (Fig.1 - 5) at the stock culture growing conditions, average present day conditions in the Chatham Rise area, and the predicted future conditions (2100) of Chatham Rise, respectively.

** The percentage changes were calculated as the changes caused by each environmental driver under the future predicted condition relative to that under the present day condition.

[*]Numbers in bold indicate statistically significant difference between the range of present day and future conditions (nitrate treatments: 6.0 and 12.0 μM; phosphate treatments: 0.4 and 2 μM; irradiance treatments: 80 and 190 μmol photons m$^{-2}$ s$^{-1}$; temperature treatments: 11°C, 15°C and 20°C) based on the one-way ANOVA. "n.s." indicates non-significant difference (one-way ANOVA) among all the treatments used for the fitting.