# Peer review of "Environmental controls on the elemental composition of a Southern Hemisphere strain of the coccolithophore *Emiliania huxleyi"

_Biogeosciences, 2017_

## Referee Comment (RC1) · L.T. Bach (Referee) · 11 Sep 2017

Review on: "Environmental controls on the elemental composition of a Southern Hemisphere strain of the coccolithophore Emiliania huxleyi " by Feng et al. In this study, Feng et al., investigate the response of E. huxleyi (cell quota, organic, inorganic matter ratios) to different environmental drivers. Their findings are very interesting and relevant in the context of phytoplankton physiology under future ocean conditions. I have, however, one major and several additional concerns that should be addressed before their study is published. I hope this review helps to improve the manuscript.

MAJOR COMMENTS:

[Figure]

Results: I am concerned about the "cellular content response (POC, PON, POP) to environmental drivers". Organic matter quotas are strongly determined by the cell cycle. POC/cell, for example, will be much lower directly after cell division than right before. Thus, you can only compare cell quotas among treatments, when you are sure that all treatments were in the same cell cycle stage. The Authors do not indicate if samples were taken at the same time. This information would be a step forward because it could then at least be assumed that cell division was synchronized during night. However, even if sampling times were identical, it remains questionable if this assumption is valid for every treatment. Growth rates are not reported here but I assume that they are below $\mu$=0.69, at least in some treatments (e.g. the low temperature treatment). If the cells divide less than once per day and only divide during night, it means that some cells of the population are packed with POC while others are depleted. Since you only sample once at an unknown cell cycle stage, it may become difficult (if not impossible) to disentangle the cell cycle-specific response from the actual treatment response. I therefore think that the results on cell quotas presented here (but also elsewhere in the literature) could potentially be misleading.

My suggestion would be to show production rates ($\mu$ x cell quota) instead of cell quotas. These also have theoretical issues but should more robust.

ADDITIONAL COMMENTS:

Page 2 Line 1: the "each" could probably be removed.

Page 2 Line 1: perhaps remove "cellular" because PIC is extracellular.

Page 2 Line 2: "implications for coccolithophore biogeochemistry". This is a rather vague formulation. What is coccolithophore biogeochemistry? Do you mean the influence coccolithophores have on biogeochemical cycles? I think a bit more precision would improve the final sentence? Do you mean their influence on the nutrient cycle? Carbon export?

Section 2.1 provides a thorough description of the culturing methodology. One crucial information should be added, however. Were all samples taken at the same time (within an appropriate time window, e.g. ∼2 hours)? This is important because cell quotas change over the day and these can only be compared when all treatments were in the same cell cycle state when sampled (see also MAJOR COMMENT).

Page 6 Line 5: Agreed but a reference for this statement would probably be useful.

Page 12 Line 10: "with lowest nitrate and phosphate concentrations of 3.6 and 0.4 $\mu$M, respectively". In this case, your results may not really be comparable to Paasche's and others. Your nutrient concentrations were not leading to zero growth whereas those of Paasche et al were.

Page 13 Line 26: check spelling of "cell".

Page 13 Line 26: It is unclear in this sentence whether you measured cell size or you refer to earlier results. Please clarify.

Page 15 Line 1: This final speculation in the temperature section is a bit too extreme. It became clearer during the last couple of years that extrapolations from the (mono-clonal) bottle to the global ocean should be avoided since way too many factors (e.g. ecology) are neglected.

Page 15 Line 9: "In general, cell growth of E. huxleyi is less limited by low CO2 concentrations than in other phytoplankton groups (Clark and Flynn, 2000; Paasche et al., 1996; Riebesell et al., 2000a)." This statement implies that E. huxleyi would have a particularly efficient CCM but is this supported by the evidence provided in the cited references? I suggest to check the MIMS-based papers by for example Björn Rost's group because these provide K1/2 values for carbon uptake and they have investigated quite a number of different species that can be compared with E. huxleyi.

Page 15 Line 27: I do not understand where the "both" is referring to.

Page 15 Line 27: "ecological implications"? Do you mean "biogeochemical implications"?

Page 16 Line 2: Confusion: The 14C measurement is not referring to your study, or is it? 14C measurements have not been described in the methods or did I miss something?

Page 16 Line 5: Reference missing in the reference list. (Please check the entire list since there some others missing as well).

Page 16 Line 25: Semicolon

Page 17 Line 16: "...future research on a full environmental matrix is still necessary." It would be valuable to add that the goal of such a matrix should not be to simply combine different factors and then use the outcome to extrapolate it to the future. The goal of culture studies should be to understand the underlying mechanisms of synergistic effects. For example: "How does the light intensity modify the temperature response and why?"

Page 18 Lines 14-17: I am not so sure about the final conclusion and the concomitant suggestion. If we design experiments to mimic anticipated physico-chemical conditions of the future as close as possible than the results can in most cases only be used to project findings from a culture experiment to the global ocean in a one to one manner. This, however, is questionable since many factors in that can significantly modify the outcomes are neglected in the experiment. Perhaps it may be more sustainable to suggest that experimentalists should design experiments in such a way that underlying mechanisms for synergistic effects can be understood.

Figure 7: Perhaps rather call it conceptual figure. Furthermore, were abbreviations "Q" defined in the text?

With kind regards, Lennart Bach

---

## Referee Comment (RC2) · Anonymous Referee #2 · 13 Sep 2017

This manuscript presents data examining how environmental conditions control the elemental composition (C, N, P) in a Southern Ocean isolate of the coccolithophore Emiliania huxleyi. The authors thoroughly test a wide assortment of environmental parameters using laboratory batch culturing experiments. This manuscript provides additional data that expands upon a recently published paper by the authors (Feng et al., 2017). Overall the manuscript is easy to follow, though there are a number of typos and inconsistencies that need to be addressed in the text and tables. The gradients across which the authors assess elemental composition are extensive, though regrettably there is no exploration of synergistic relationships between the variables (which the authors acknowledge). The authors state in the text that they were able to rank the

importance of the different environmental variables, but the table containing that information (Table 3) was not included in the manuscript pdf, making it difficult to comment on that topic.

Specific comments:

Not sure if this is a journal formatting issue but there should be either spaces or indentation to separate paragraphs. This is consistent throughout the manuscript.

Inconsistent use of serial commas

The authors are inconsistent in using the modifier "cellular" when referring to the various forms of particulate organic matter. If, as I suspect, they are only referring to cellular forms of such matter, then the continual use of the "cellular" term is unnecessary.

Pg7 line7: Delete 'then'.

Figures 1-5: If you are fitting curves through data points, would it not be better to plot all of your data points using a scatterplot as opposed to using bar plots? This would give the reader a much better sense of the variability within the data.

Section 3.1: You don't mention anything about the effects of nutrients on POC.

Why are the values for goodness of fit in the supplement and not in the manuscript?

Pg9, line 15: The 'dramatic' decline was predominately seen between 4 and 7°C and leveled out thereafter. Maybe change the wording to more correctly state this response.

Table2: The meaning of bold values should be stated in the table caption, not in the manuscript text.

Table2: There are numerous values that are stated as being significantly different in the text but are not bold in the table.

Table2: Why are these data presented as a table instead of plots as were used for the

previous metrics?

Section 3.8: I could not find the Table 3 that is referenced in the text, making it difficult to review this section.

Pg13, line26: typo

Pg13, line27: typo

Pg14, line2: typo

Pg14, line5: Why are cell size data not presented (in text or supplement) in either this manuscript or Feng et al., 2017?

Pg14, line8: Don't you mean greater than 11°C, since 10°C was not tested in this study and PIC values did not appear to differ amongst the 4, 7, and 11°C treatments?

Pg14, line10: Again referring to data (cell volume) that is not presented.

Pg14, line11: 10°C was not a treatment level in this study.

Pg14, line15: The best-fit line does not follow this description. Given the poor fit based on the low R2 value, why is this fitting included?

Pg14, line16: 24°C was not a temperature used in this study or Feng et al. (2017)

Pg14, line23: A 74% increase is not really 'almost double'.

Pg16, line2: This study did not use any isotopic labeling. I assume that this is referring to Feng et al. (2017).

Pg17, line23: You could also cite Blanco-Ameijeiras et al. (2016) in PLoS ONE since they tested 13 strains under the same environmental conditions, avoiding inter-laboratory experimental variability that is an issue when comparing results from different experiments.

---

## Short Comment (SC1) · 15 Sep 2017

This work on Ehux elemental composition provides a valuable companion to the physiological parameters (growth, photosynthetic and calcification rates) from the very same experiments, as already published in Limnol., Oceanogr. 62, 519-540 (2017). While the work is limited to a single New Zealand Ehux culture strain, isolated in 2009, the overall experimental set-up of comparing the impact 5 environmental drivers (on their own, with all other drivers kept at those of the stock culture conditions) is interesting. The limitations of the work in predicting global Ehux climate responses are carefully defined.

---

## Author Comment (AC1) · 13 Oct 2017

Dear reviewer and editors,

The authors appreciate the constructive comments from the reviewer very much. According to the reviewer's comments, the manuscript has been thoroughly revised. The responses to the reviewer's comments are listed below.

Response to the general comments: "Overall the manuscript is easy to follow, though there are a number of typos and inconsistencies that need to be addressed in the text and tables. The gradients across which the authors assess elemental composition

[Figure]

are extensive, though regrettably there is no exploration of synergistic relationships between the variables (which the authors acknowledge). The authors state in the text that they were able to rank the importance of the different environmental variables, but the table containing that information (Table 3) was not included in the manuscript pdf, making it difficult to comment on that topic." The typos and the inconsistencies have been carefully checked and fixed. The potential synergistic relationships between the variables have now been further explored. The missing Table 3 is also added in the revised version.

Response to the specific comments: "Not sure if this is a journal formatting issue but there should be either spaces or indentation to separate paragraphs. This is consistent throughout the manuscript." Spaces have been added to separate paragraphs.

"Inconsistent use of serial commas" This problem has been fixed. The serial commas are now consistently used.

"The authors are inconsistent in using the modifier "cellular" when referring to the various forms of particulate organic matter. If, as I suspect, they are only referring to cellular forms of such matter, then the continual use of the "cellular" term is unnecessary." In this manuscript, the elemental composition mainly refers to the cellular elemental contents and ratios, and the inconsistent use of the modifier "cellular" is now fixed.

"Pg7 line7: Delete 'then'." The word "then" has been deleted.

"Figures 1-5: If you are fitting curves through data points, would it not be better to plot all of your data points using a scatterplot as opposed to using bar plots? This would give the reader a much better sense of the variability within the data." The bar plots instead of scatter plots are used in order to make comparisons between different treatments for each of the manipulation experiments. And the fitting curves are used to describe the variability of the trends within the data.

"Section 3.1: You don't mention anything about the effects of nutrients on POC." There

were no significant effects of nutrient concentrations on the POC contents based on our experimental results.

"Why are the values for goodness of fit in the supplement and not in the manuscript?" There are 7 figures and 3 tables in the manuscript. Therefore, the table containing the fitting equations and the values for goodness of fit are in the supplement in order to keep the manuscript to a reasonable length.

"Pg9, line 15: The 'dramatic' decline was predominately seen between 4 and 7°C and leveled out thereafter. Maybe change the wording to more correctly state this response." The wording has now been changed to "The C:Chl-a ratio dramatically decreased with warming, especially between 4°C and 7°C (Fig. 6d)".

"Table 2: The meaning of bold values should be stated in the table caption, not in the manuscript text. Table 2: There are numerous values that are stated as being significantly different in the text but are not bold in the table." The meaning of bold values has been stated in the table caption in the revised manuscript. The other significantly different values in the table have also been formatted in bold font.

"Table2: Why are these data presented as a table instead of plots as were used for the previous metrics?" The differences of the elemental ratios between different treatments are less significant compared to the cellular elemental contents. Therefore, the ratios are presented in one table and not in 3 separate figures, to keep a reasonable total number of figures of the manuscript.

"Section 3.8: I could not find the Table 3 that is referenced in the text, making it difficult to review this section." The table has been added in the manuscript.

"Pg13, line26: typo; Pg13, line27: typo; Pg14, line2: typo" These typos have all been corrected.

"Pg14, line5: Why are cell size data not presented (in text or supplement) in either this manuscript or Feng et al., 2017?" The cell size data from the temperature manipulation

experiments is presented in the supplements as Fig. S1.

"Pg14, line8: Don't you mean greater than 11°C, since 10°C was not tested in this study and PIC values did not appear to differ amongst the 4, 7, and 11°C treatments? The previous 10°C is now revised to 11°C.

Pg14, line10: Again referring to data (cell volume) that is not presented. Now the supplemental data in Fig. S1 is referred to.

Pg14, line11: 10°C was not a treatment level in this study. The previous 10°C is now revised to 11°C.

"Pg14, line15: The best-fit line does not follow this description. Given the poor fit based on the low R2 value, why is this fitting included?" The PIC:POC ratio at 4°C was significantly lower than the other treatments, indicating lower cellular PIC:POC production under extreme low temperature; therefore, this fitting is included.

"Pg14, line16: 24°C was not a temperature used in this study or Feng et al. (2017)" The authors agree that 24°C was not a temperature used in the experiment; however, here 24°C was the optimal temperature for photosynthetic rate from the fitting in Fig. 3d of Feng et al. (2017).

"Pg14, line23: A 74% increase is not really 'almost double'." The original wording of "…almost double…" has been revised to "the cellular N:P ratio of E. huxleyi at 20°C increased by 74%...".

"Pg16, line2: This study did not use any isotopic labeling. I assume that this is referring to Feng et al. (2017)." Yes, this is referring to Feng et al. (2017). And the reference has been added in the text.

"Pg17, line23: You could also cite Blanco-Ameijeiras et al. (2016) in PLoS ONE since they tested 13 strains under the same environmental conditions, avoiding interlaboratory experimental variability that is an issue when comparing results from different experiments." The reference of Blanco-Ameijerias et al. (2016) has been cited.

We look forward to hearing back from you again. Thank you very much.

Sincerely,

Yuanyuan Feng and the coauthors

Please also note the supplement to this comment:
https://www.biogeosciences-discuss.net/bg-2017-332/bg-2017-332-AC1-supplement.pdf

---

## Author Comment (AC2) · 13 Oct 2017

Dear reviewer and editors,

The authors would like to thank the reviewer for the helpful comments provided in order to improve the manuscript. The manuscript has been carefully checked and revised based on the reviewer's comments. The responses to the detailed comments are listed below.

"MAJOR COMMENTS: Results: I am concerned about the "cellular content response (POC, PON, POP) to environmental drivers". Organic matter quotas are strongly determined by the cell cycle. POC/cell, for example, will be much lower directly after cell division than right before. Thus, you can only compare cell quotas among treatments, when you are sure that all treatments were in the same cell cycle stage. The Authors do not indicate if samples were taken at the same time. This information would be a step forward because it could then at least be assumed that cell division was synchronized during night. However, even if sampling times were identical, it remains questionable if this assumption is valid for every treatment. Growth rates are not reported here but I assume that they are below 0.69, at least in some treatments (e.g. the low temperature treatment). If the cells divide less than once per day and only divide during night, it means that some cells of the population are packed with POC while others are depleted. Since you only sample once at an unknown cell cycle stage, it may become difficult (if not impossible) to disentangle the cell cycle-specific response from the actual treatment response. I therefore think that the results on cell quotas presented here (but also elsewhere in the literature) could potentially be misleading. My suggestion would be to show production rates ($\mu$ x cell quota) instead of cell quotas. These also have theoretical issues but should more robust. " The authors agree that the cellular elemental quotas depend on the cell cycles. The samples in our study are taken during the same time window so that most of the cells were in the same stage during sampling. The information was also added in the results section as "Samples were collected for cell counts, Chl-a biomass, and elemental components, including particulate organic carbon (POC), particulate inorganic carbon (PIC), particulate organic nitrogen (PON), and particulate organic phosphorus (POP), starting 2 hours after the beginning of the light incubation phase and finishing within 2 hours for all the experimental treatments". In addition, the cells were examined under the microscope; there were no significantly enlarged cells in division observed, even at the lowest temperature. Therefore, the results of the elemental compositions presented in our study are comparable among different treatments.

ADDITIONAL COMMENTS: "Page 2 Line 1: the "each" could probably be removed." The word "each" has been removed.

[Figure]

"Page 2 Line 1: perhaps remove "cellular" because PIC is extracellular." The word "cellular" has been removed.

"Page 2 Line 2: "implications for coccolithophore biogeochemistry". This is a rather vague formulation. What is coccolithophore biogeochemistry? Do you mean the in-fluence coccolithophores have on biogeochemical cycles? I think a bit more precision would improve the final sentence? Do you mean their influence on the nutrient cycle? Carbon export?" The text has been revised to "...with wide-reaching implications for coccolithophore related marine biogeochemical cycles...".

"Section 2.1 provides a thorough description of the culturing methodology. One crucial information should be added, however. Were all samples taken at the same time (within an appropriate time window, e.g. ~2 hours)? This is important because cell quotas change over the day and these can only be compared when all treatments were in the same cell cycle state when sampled (see also MAJOR COMMENT)." All the samples from each manipulation experiments were collected in a similar and appropriate time window. This description has been added in the results section as stated above.

"Page 6 Line 5: Agreed but a reference for this statement would probably be useful." A reference has been added.

"Page 12 Line 10: "with lowest nitrate and phosphate concentrations of 3.6 and 0.4 $\mu$M, respectively". In this case, your results may not really be comparable to Paasche's and others. Your nutrient concentrations were not leading to zero growth whereas those of Paasche et al were." The authors agree that the lowest nutrient concentrations were not as low (leading to zero) as those in the cited references. However, here we made the comparisons only to point out the difference of the results we observed in our study and those under nutrient depleted conditions. It is stated in the manuscript that "the present study used a semi-continuous incubation method with higher and relatively steady nutrient concentrations (with lowest nitrate and phosphate concentrations of 3.6 and 0.4 $\mu$M, respectively) and the cells were grown and sampled at a healthy exponential growth phase". And thus "further studies at extremely low nutrient concentrations (<0.1 $\mu$M) in a steady-state growth phase are still needed to understand the potential connection between carbon production and extreme nutrient limitation".

"Page 13 Line 26: check spelling of 'cell'." The original typo has been corrected.

"Page 13 Line 26: It is unclear in this sentence whether you measured cell size or you refer to earlier results. Please clarify." The new supplemental figure (Fig. S1) has been added in order to provide the cell size information from the temperature experiments.

"Page 15 Line 1: This final speculation in the temperature section is a bit too extreme. It became clearer during the last couple of years that extrapolations from the (monoclonal) bottle to the global ocean should be avoided since way too many factors (e.g. ecology) are neglected." The last sentence has been revised to "Similarly, Toseland et al. (2013) suggested that future warming might accentuate nitrate limitation in the oceans" to avoid over extrapolations from our bottle incubation experiments.

"Page 15 Line 9: 'In general, cell growth of E. huxleyi is less limited by low CO2 concentrations than in other phytoplankton groups (Clark and Flynn, 2000; Paasche et al., 1996; Riebesell et al., 2000a).' This statement implies that E. huxleyi would have a particularly efficient CCM but is this supported by the evidence provided in the cited references? I suggest to check the MIMS-based papers by for example Björn Rost's group because these provide K1/2 values for carbon uptake and they have investigated quite a number of different species that can be compared with E. huxleyi." The reference of Rost et al. (2003) that examined the K1/2 values for carbon uptake of several phytoplankton species is now cited in the revised manuscript.

"Page 15 Line 27: I do not understand where the 'both' is referring to." The word "both" refers to the two parameters 1. cellular PIC:POC ratio in the present manuscript and 2. the ratio of calcification rate vs. photosynthesis rate in Feng et al. (2017) being commonly used in research papers to indicate the relative change of PIC vs. POC production in coccolithophores, and thus they have implications for the marine rain

ratio.

"Page 15 Line 27: 'ecological implications'? Do you mean 'biogeochemical implications'?" The word "ecological" has been revised to "biogeochemical".

"Page 16 Line 2: Confusion: The 14C measurement is not referring to your study, or is it? 14C measurements have not been described in the methods or did I miss something?" The paper Feng et al. (2017) is now referred to in the text.

"Page 16 Line 5: Reference missing in the reference list. (Please check the entire list since there some others missing as well)." The missing references are thoroughly checked and added in the reference list.

"Page 16 Line 25: Semicolon" The semicolon has been changed to comma.

"Page 17 Line 16: '. . .future research on a full environmental matrix is still necessary.' It would be valuable to add that the goal of such a matrix should not be to simply combine different factors and then use the outcome to extrapolate it to the future. The goal of culture studies should be to understand the underlying mechanisms of synergistic effects. For example: 'How does the light intensity modify the temperature response and why?' " The goal of these research on full environmental matrix has been added in the revised manuscript as: "These experiments will not only help to further explore the potential interactions (i.e. synergistic or agnostic effects) between environmental drivers, but also provide a better understanding of the underlying mechanisms of these interactive effects".

"Page 18 Lines 14-17: I am not so sure about the final conclusion and the concomitant suggestion. If we design experiments to mimic anticipated physico-chemical conditions of the future as close as possible than the results can in most cases only be used to project findings from a culture experiment to the global ocean in a one to one manner. This, however, is questionable since many factors in that can significantly modify the outcomes are neglected in the experiment. Perhaps it may be more sustainable to

suggest that experimentalists should design experiments in such a way that underlying mechanisms for synergistic effects can be understood." The authors agree that there are many factors in the oceanic environments neglected in our experiment. However, it is a general limitation of laboratory manipulation experiments. Our manipulation experiments focused on the single driver effects, which provide some helpful diagnostic information for further explaining the interactive effects of multiple drivers. As such, the final conclusions have been further extended as: "For future multi-factorial manipulation experimental designs, our results suggest that the magnitudes of change in each environmental driver need to be determined/decided cautiously and should have environmental relevance in order to make more accurate predictions, and the understanding of interactive effects of multiple environmental drivers and the underlying mechanisms should be further explored.".

"Figure 7: Perhaps rather call it conceptual figure. Furthermore, were abbreviations "Q" defined in the text?" The figure legend has been changed to "conceptual figure", and the abbreviation of "Q" has also been defined as cellular quota.

We look forward to hearing back from you again. Thank you very much.

Sincerely, Yuanyuan Feng and the coauthors

Please also note the supplement to this comment:
https://www.biogeosciences-discuss.net/bg-2017-332/bg-2017-332-AC2-supplement.pdf

———————————————

---

## Author Comment (AC3) · 13 Oct 2017

The revised manuscript will be uploaded separately.
* * *

---

## Author Comment (AC4) · 13 Oct 2017

The revised manuscript will be uploaded seperately.

---

## Author Comment (AC5) · 13 Oct 2017

[revised manuscript text omitted]
- 20 Press, Cambridge, United Kingdom and New York, NY, USA, 2013.
  Stojkovic, S., Beardall, J., and Matear, R.: CO2 concentrating mechanisms in three southern hemisphere strains of *Emiliania huxleyi*, J. Phycol., 49, 670-679, 2013.
  Toseland, A., Daines, S. J., Clark, J. R., Kirkham, A., Strauss, J., Uhlig, C., Lenton, T. M., Valentin, K., Pearson, G. A., Moulton, V., and Mock, T.: The impact of temperature on marine phytoplankton
- 25 resource allocation and metabolism, Nat. Clim. Chang., 3, 979-984, doi:10.1038/nclimate1989, 2013. van Rijssel, M., and Gieskes, W. W. C.: Temperature, light, and the dimethylsulfoniopropionate (DMSP) content of *Emiliania huxleyi* (Prymnesiophyceae), J. Sea Res., 48, 17-27, doi:10.1016/s1385-1101(02)00134-x, 2002.
  - 26

Watabe, N., and Wilbur, K. M.: Effects of temperature on growth calcification and coccolith form in *Coccolithus huxleyi* (coccolithineae), Limnol. Oceanogr., 11, 567-575, 1966.

Westbroek, P., Brown, C. W., Vanbleijswijk, J., Brownlee, C., Brummer, G. J., Conte, M., Egge, J., Fernandez, E., Jordan, R., Knappertsbusch, M., Stefels, J., Veldhuis, M., Vanderwal, P., and Young, J.: A

5 model system approach to biological climate forcing - the example of *Emiliania huxleyi*, Global Planet. Change, 8, 27-46, doi:10.1016/0921-8181(93)90061-r, 1993.
Welschmeyer, N. A. Fluorometric analysis of chlorophyll *a* in the presence of chlorophyll *b* and

pheopigments. Limnol. Oceanogr. 39, 1985–1992, 1994.

Young, J. R., Poulton, A. J., and Tyrrell, T.: Morphology of Emiliania huxleyi coccoliths on the North

West European shelf - is there an influence of carbonate chemistry?, Biogeosciences Discussions, 11, 4531-4561, 2014.

Zondervan, I., Zeebe, R. E., Rost, B., and Riebesell, U.: Decreasing marine biogenic calcification: A negative feedback on rising atmospheric  $pCO_2$ , Global Biogeochem. Cy., 15, 507-516, doi:10.1029/2000gb001321, 2001.

15 Zondervan, I., Rost, B., and Riebesell, U.: Effect of CO2 concentration on the PIC/POC ratio in the coccolithophore *Emiliania huxleyi* grown under light-limiting conditions and different daylengths, J. Exp. Mar. Biol. Ecol., 272, 55-70, doi:10.1016/s0022-0981(02)00037-0, 2002.

Zondervan, I.: The effects of light, macronutrients, trace metals and CO2 on the production of calcium carbonate and organic carbon in coccolithophores - a review, Deep-Sea Res. PT II, 54, 521-537, doi:10.1016/j.dsr2.2006.12.004, 2007.

---

## Author Response (AR2)

Dear reviewer and editors,

The authors appreciate the helpful comments from the reviewer very much, which help to improve the manuscript. According to these comments, the manuscript has been thoroughly revised. The responses to the reviewer's comments are listed below.

*"Page 2, Line 5: What kind of 'climate change' are you talking about? "*

The previous wording of "climate change" has been revised to "The global climate changed induced by anthropogenic activities"

*"Page 4, Lines 5-6: "The marine coccolithophore Emiliania huxleyi (morphotype A, strain NIWA1108) was isolated from the Chatham Rise, east of New Zealand by Dr. H. Chang in 2009, as detailed in Feng et al. (2017).*

*I have checked Feng et al. (2017) and Feng (2014) (PhD thesis of the corresponding author of this manuscript that shows original study of this manuscript), however, could not find detailed information of the strain in the papers. Please refer an appropriate paper that describes detailed information of the strain, or provide following information in this manuscript instead; sampling locality (latitude and longitude), sampling method, depth of seawater sample that yielded NIWA1108, hydrographic condition (temperature, salinity, etc.) of seawater at the sampling time, date of sampling, isolation method, and information of sampling cruise and cruise report. There is a possibility that culture strains originate from different hydrographic condition have different environmental preferences, as authors mention "we can speculate that the diverse E. huxleyi strains growing in different temperature regions might have different requirements for nitrogen ...." in the page 14. I think the detailed information of origin culture strain will help discussion of this study, and comparison of results from this study with result of other studies."*

The detailed information of the *Emiliania huxleyi* strain NIWA 1108 has been added in the revised manuscript as "The marine coccolithophore *Emiliania huxleyi* (morphotype A, strain NIWA1108) was isolated from the surface water (depth of 5-6 m, salinity of 34.78) at 41°35.8'S 175°41.5'E, east of New Zealand by Dr. H. Chang aboard the *RV Tangaroa* on the research voyage TAN0909 in November 2009.

The water temperature was 12.1°C at the sampling site".

*"Page 4, Line 14; Please describe how did you know the culture strain is in exponential growth phase."*
The text "determined by the growth curve" has been added in order to clarify how the exponential growth phase was determined.

*"Page 5, Line 18: How long did you keep your subsample in dark 4˚C condition before counting?"*
The text has been revised to "…and stored in dark at 4°C for no more than 5 days before counting".

*"Page 11, Line 15: Put 'E. huxleyi' in italics."*
The font has been corrected.

*"Page 13, Lines 14-16: "In our 15 accompanying study, the growth, photosynthetic, and calcification rates all increased with rising temperature until the optimal temperature was reached (Feng et al., 2017)";*
*Please mention the optimal temperature of the strain NIWA 1108 reported by Feng et al. (2017), and then discuss whether the optimal temperature of the strain in laboratory experiment is same/close to the temperature of sampling locality."*
The optimal temperatures for growth, photosynthetic and calcification rates are now included and discussed in the revised manuscript, as "…the growth, photosynthetic, and calcification rates all increased with rising temperature until the optimal temperature was reached at 25°C, 24°C, and 20°C respectively (Feng et al., 2017), which were all higher than the stock culture growth temperature or the temperature at the isolation site of *E. huxleyi* strain NIWA 1108".

Thank you again for your help in the review process of the manuscript. Look forward to hearing back from you soon.

Sincerely yours,
Yuanyuan Feng and the co-authors

[revised manuscript text omitted]

---

## Author Response (AR3)

Dear editors,

It is a great honor to know that the manuscript is now accepted for publication in your journal. Thank you very much for all the efforts of yours and the reviewers to improve the manuscript.

The final version of the manuscript has been revised according to your comments. The point-to-point response is listed below.

*"P1, L19: nitrate and phosphate concentrations"*
The change has been made accordingly.

*P1, L20: partial pressure of CO2 (pCO2)*
The change has been made accordingly.

*P3, L15–16: nutrient level*
The wording of "nutrient concentration" has been revised to "nutrient level".

*P3, L22: the magnitude of changes in each environmental driver*
The word "of" has been changed to "in".

*P5, L8: Add a space between ± and 0.*
The space has been added.

*P11, L22: "E. huxleyi" should be italic.*
"E. huxleyi" is now italic.

Thank you again and look forward to hearing back from you soon.

Sincerely yours,
Yuanyuan Feng and the co-authors